# Provably Efficient Reward Transfer in Reinforcement Learning with Discrete Markov Decision Processes

**Kevin Vora** *kvora1@asu.edu*
*School of Computing and Augmented Intelligence*
*Arizona State University*

**Yu Zhang** *yzhan442@asu.edu*
*School of computing and Augmented Intelligence*
*Arizona State University*

**Reviewed on OpenReview:** *https://openreview.net/forum?id=u2b31c9Noe*

## Abstract

In this paper, we propose a new solution to reward adaptation (RA) in reinforcement learning, where the agent adapts to a target reward function based on one or more existing source behaviors learned a priori under the same domain dynamics but different reward functions. While learning the target behavior from scratch is possible, it is often inefficient given the available source behaviors. Our work introduces a new approach to RA through the manipulation of Q-functions. Assuming the target reward function is a known function of the source reward functions, we compute bounds on the Q-function and present an iterative process (akin to value iteration) to tighten these bounds. The iteration process is based on a lite-model, which is assumed to be given or can be learned. The computed bounds enable action pruning in the target domain before learning even starts. We refer to this method as "*Q-Manipulation*" (Q-M). We formally prove that Q-M, under discrete domains and an accurate lite-model, does not affect the optimality of the returned policy and show that it is provably efficient in terms of sample complexity. Q-M is evaluated in a variety of synthetic and simulation domains to demonstrate its effectiveness, generalizability, and practicality.

## 1 Introduction

Reinforcement Learning (RL) as described by Watkins (1989); Sutton & Barto (2018) represents a class of learning methods that allow agents to learn from interacting with the environment. RL has demonstrated great successes in various domains such as games like Chess in Campbell et al. (2002), Go in Silver et al. (2016), and Atari games in Mnih et al. (2015), logistics in Yan et al. (2022), biology in Angermueller et al. (2019), and robotics in Kober et al. (2013). However, applying RL to many real-world problems still suffers from the issue of high sample complexity. Prior approaches have been proposed to alleviate the issue from different perspectives, such as learning optimization, transfer learning, modular and hierarchical RL, and offline RL. However, few methods provably improve sample complexity.

The problem of reward adaptation (RA) was first introduced and addressed by Barreto et al. (2017; 2020), where the learning agent adapts to a target reward function given one or multiple existing behaviors learned a priori (referred to as the source behaviors) under the same actions and transition dynamics but different reward functions. RA has many useful applications, such as enabling a vehicle's driving behavior from two known behaviors (comfortable driving with passengers and fast driving for goods delivery) to a new target behavior that combines comfort and speed, accommodating both passengers and goods. Featuring such a special type of transfer learning, RA methods can benefit from an ever-growing repertoire of source behaviors to create new and potentially more complex target behaviors. Learning the target behavior from scratch via

model-free methods is possible but often inefficient given the available source behaviors, while model-based methods must implement bookkeeping to merge sample information from different source behaviors since it requires sufficient samples for every part of the state space, which can be expensive. In this paper, we present a new approach that takes the advantage of both sides while avoiding their limitations to bridge the gap for RA. This lite-model offers complimentary benefits with respect to the previous work.

To better conceptualize the RA problem, consider a grid-world as shown in Fig. 1, which is an expansion of the Dollar-Euro domain described by Russell & Zimdars (2003). In this domain, the agent can move to any of its adjacent locations at any step. The agent's initial location is colored in yellow, and the terminal locations are colored pink or green, which correspond to the source reward functions (i.e., collecting dollars and euros), respectively. Visiting the terminal location with a single color returns a reward of 1.0 under the corresponding reward function, and visiting the terminal location with split colors returns a reward of 0.6 under both reward functions. A target domain may correspond to a reward function that awards both dollars and euros.

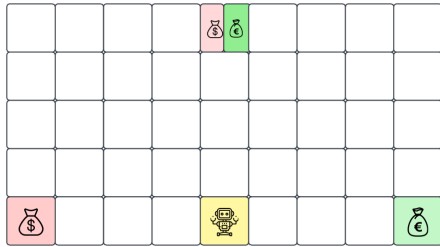

Figure 1: Dollar-Euro domain.

RA is most efficient when the source and target domains feature reward functions that are correlated. For example, an assumption was made in Successor Feature Q-Learning (SFQL) Barreto et al. (2017; 2020) where the reward functions are expressed as feature weights. An advantage of this assumption is that the source behaviors can be evaluated easily under the target domain. SFQL can be viewed as combining the best parts of the source behaviors to initialize learning. Consequently, SFQL may not work well for situations where the target behavior differs substantially from the source behaviors, such as in the Dollar-Euro domain. Our proposed approach, on the other hand, represents a more general knowledge transfer method whose efficacy does not rely on the similarity between the source and target behaviors. However, we do assume that training of the source behaviors jointly covers the state space. Soft Q Bounding (SQB) Adamczyk et al. (2024) establishes double sided bounds of the Q function to speed up learning by clipping overly optimistic or pessimistic updates. When viewed as a method for RA, it relies on a given Q function computed from source behaviors. Although the proposed bounds are related to Q-M bounds, the ways we are estimating and using these bounds differ substantially.

Our approach to RA is referred to as "*Q-Manipulation*" (Q-M). In this paper, we focus on discrete Markov Decision Processes (MDPs) for a theoretical treatment. Challenges in addressing continuous MDPs are discussed in Section 5, with directions outlined for future work. Similar to prior work on RA, we assume the relationship between the source reward functions and the target reward function is known, and in our case, via what is referred to as a *combination function*. The intuition here is that we often have a good idea about the functional relationship (potentially noisy) between the source and target reward functions, e.g., being linear in the Dollar-Euro domain. Based on such a relationship, Q-M computes an upper and lower bound of the Q-function in the target domain via an iterative process similar to value iteration. This process operates on a lite-model of the transition function that is assumed to be provided and is accurate. To learn such a model accurately in practice when the model is not available, an assumption must be made that each transition is experienced at least once, which is a less stringent requirement than that of value convergence of RL. The output bounds enable us to prune actions before learning the target behavior, without affecting its optimality. In our evaluation, we empirically show that the effectiveness of Q-M across simulated and randomly generated domains, and also analyze its limitations. In general, Q-M operates under additional computation and space requirements that are reasonable to implement in practice, with its benefits to sample complexity outweighing these costs. For a comprehensive comparison between Q-M and competing transfer learning approaches, refer to Table 1.

Our core contributions are: We address the problem of reward adaptation (RA) via Q-Manipulation (Q-M) in domains with discrete state and action spaces, and demonstrate that Q-M is provably efficient to sample complexity due to action pruning, and represents a new approach to RA that supports more general knowledge transfer than the previous work. We introduce two methods that both leverage a lite-model for capturing neighboring-state information: 1) Q-Manipulation (Q-M), which modifies Bellman updates to compute

upper and lower bounds on the target domain's $Q^*$ in a value-iteration style update and 2) Monotonic Q-Manipulation (M-Q-M), which extends Q-M by incorporating source Q-functions for initializing the bounds and then iteratively tighten them. We formally prove the correctness of both methods. Since optimal actions are preserved , there is no negative transfer under the ideal setting of accurate lite-models and Q functions (in the case of M-Q-M). We extensively evaluate Q-M against baselines under such an ideal setting and with a more realistic setting where the lite-model and Q functions are learned to validate its efficacy and analyze its limitations.We also investigate under linear, nonlinear, and imperfect combination functions. Results confirm Q-M as a valuable approach for RA.

## 2 Related work

Transfer learning and multi-task learning have emerged as two central paradigms in RL for improving sample efficiency and generalization. This section surveys existing approaches across these domains, with a focus on how prior experience from one or more source tasks can be exploited to accelerate or stabilize learning the target task.

**Transfer Learning in Reinforcement Learning:** The goal of transfer learning in RL is to utilize knowledge gained from previously solved tasks (source tasks) to improve performance in a different, typically unseen task (target task). Foundational surveys such as Taylor & Stone (2009); Wulfmeier et al. (2023) categorize transfer techniques by how the knowledge is transferred, ranging from policies and value functions to learned models and internal representations. According to the taxonomy of transfer reinforcement learning proposed by Taylor & Stone (2009), reward adaptation belongs to transfer learning where the allowed task difference is $R$. Even though more general transfer methods can be applied to reward adaptation, such as Mann & Choe (2013), these methods are often heuristic in nature due to the generic relationship between the source and target domains. Within the category of reward adaptation in transfer RL, several relevant methods have been proposed. Notably, SFQL and SQB represent recent efforts aimed at improving transfer efficiency through the use of successor features ($\phi$ and $w$) and action-value bounds, respectively. Table 1 presents a comprehensive comparison of three methods, detailing their key characteristics, including assumptions, core strategies, and intended use cases to provide a clearer understanding of the purpose and design of each approach.

| | | Q-M (M-Q-M) | SFQL Barreto et al. (2017) | SBQ Adamczyk et al. (2024) |
|---|---|---|---|---|
| **Assumptions** | Source Domain | $R_i$, $\hat{T}(\cdot \mid s, a)$, Q-variants (M-Q-M only) | $\phi$ and $w_i$ | Initial Q |
| | Target Domain | $f$ | $w_{target}$ | NA |
| **Strategy for transfer** | | Action pruning | Warm start | Clipped Bellman update |
| **Guaranteed improvement on sample efficiency** | | **Yes** | No | No |
| **Pre-Learning or Online Learning** | | Pre-Learning | Pre-Learning | Online Learning |
| **Modularity** | | Yes | Yes | Yes |
| **Robust against Negative Transfer** | | **Yes** | No | No |

Table 1: Comparative analysis of Q-M, SFQL and SBQ for transfer learning given source domain Q-values, target reward as combination of source reward $\mathcal{R} = f(R_i)$ and lite-model $\hat{T}(\cdot \mid s, a)$

**Multi-Task Reinforcement Learning (MTRL):** MTRL addresses transfer from a different angle by aiming to jointly learn across multiple tasks. Rather than assuming task isolation, MTRL leverages shared experience across a distribution of tasks to improve generalization or reduce training time on each task Vithayathil Varghese & Mahmoud (2020). Common methods involve parameter sharing at the level of policy networks, value functions, or representation encoders. For example, D'Eramo et al. (2019) explores decentralized training where multiple agents share parameters with a central model during training, facilitating cross-task generalization. MTRL typically involves joint learning across tasks. Moreover, MTRL approaches typically rely on strong assumptions about the alignment of tasks during training, such as tasks belonging to the same distribution Taylor & Stone (2009). These assumptions are necessary for stable joint optimization

but significantly limit robustness when attempting to generalize or transfer across structurally diverse or highly heterogeneous tasks.

Despite their promise, existing transfer RL and MTRL methods often assume a high degree of behavioral or reward similarity between source and target tasks. When this assumption does not hold, transferred knowledge can mislead the learning process, a phenomenon known as *negative transfer*. To mitigate this, some approaches investigate mechanisms for estimating or bounding task similarity prior to transfer, though such measures are not always practical, reliable, or easy to compute Carroll & Seppi (2005); Taylor & Stone (2009). The proposed approach to reward adaptation does not rely on this assumption and thus bridges an important gap in transfer learning, offering a complementary and robust strategy for leveraging prior task knowledge across tasks, regardless of their similarity.

**Other Related Paradigms:** Several adjacent fields intersect with the goals of transfer in RL and RA. Reward decomposition Russell & Zimdars (2003); Van Seijen et al. (2017) breaks down complex reward signals into composable sub-rewards to simplify learning. Multi-objective RL Roijers et al. (2013); Vamplew et al. (2011) optimizes policies that balance trade-offs among multiple predefined objectives. Hierarchical RL (HRL) Dietterich (1998); Bacon et al. (2017) structures learning around temporal abstraction, using high-level policies to control low-level sub-policies. While conceptually related, these approaches often rely on known task structure, explicit sub-goals, or carefully designed reward factorizations, which limit their applicability to general transfer scenarios or in some cases they are only scaling up RL instead of effective transfer of knowledge.

## 3 Proposed approach

In this section, we start with a brief introduction to reinforcement learning (RL) before discussing reward adaptation (RA) and formulating our approach.

### 3.1 Preliminaries

In RL, the task environment is modeled as an MDP $M = (S, A, T, R, \gamma)$, where $S$ is the state space, $A$ is the action space, $T : S \times A \times S \to [0, 1]$ is the transition function, $R : S \times A \times S \to \mathbb{R}$ is the reward function, and $\gamma$ is the discount factor. At every step $t$, the RL agent observes state $s_t$ and takes an action $a_t \in A$. As a result, the agent progresses to state $s_{t+1}$ according to the transition dynamics $T(\cdot|s_t, a_t)$, and receives a reward $r_t$. The goal is to search for a policy that maximizes the expected cumulative reward. We use $\pi : S \to A$ to denote a policy. The $Q$ function of the optimal policy $\pi^*$ is denoted by $Q^*$ and defined in Eq. 1.

To prepare us for later discussion, we also introduce $Q^\mu$ (Eq. 2) to represent the $Q$ function of the "worst" policy that minimizes the expected return. The following lemma establishes the connection between $Q^\mu$ and a variant of $Q^*$:

$$Q^*(s,a) = \max_\pi \left[ \mathbb{E} \left[ \sum_{t=0}^\infty \gamma^t r_t | s_0, \pi \right] \right] \quad (1)$$

**Lemma 3.1.** $Q^\mu_R(s,a) = -Q^*_{-R}(s,a)$, *where* $Q^*_{-R}(s,a)$ *denotes the Q function of the optimal policy under negative $R$ or $-R$.*

$$Q^\mu(s,a) = \min_\pi \left[ \mathbb{E} \left[ \sum_{t=0}^\infty \gamma^t r_t | s_0, \pi \right] \right] \quad (2)$$

In this paper, we consider RL with discrete state and action spaces and deterministic policies. Extending the discussion to the continuous cases and stochastic policies will be future work. Proofs throughout the paper are in the appendix.

### 3.1.1 Reward Adaptation (RA)

In RA, we consider only the adaptation of the reward function, and hence assume the same transition dynamics (including terminal states), state, and action spaces for the source and target behaviors. Furthermore, the source and target reward functions are assumed to be related, e.g., via successor features in Barreto et al. (2017). The source domains, $M_1, M_2, \ldots, M_n$, are no longer accessible while learning target behaviors but may *transfer knowledge* to the target domain. Next, we provide the problem statement of RA under our approach as follows:

**Definition 3.2.** Reward Adaptation: RA is a problem to determine the optimal policy under a target reward function $\mathcal{R}$ that is under a **known** functional relationship with the source reward functions specified as follows:

$$\mathcal{R} = f(R_1, R_2, \ldots R_n) \tag{3}$$

where $f$ is referred to as the combination function that relates rewards from the source and target domains.

The above definition provides a general formulation of RA when no restriction is placed on $f$, equivalent to that in Barreto et al. (2017) when each state is viewed as a unique feature. In practice, however, we often restrict the form of $f$, such as assuming linearity. In such cases, the above formulation becomes less expressive for rewards than Barreto et al. (2017) for RA problems. When $f$ must be learned, $f$ can also be augmented with a noise component to accommodate for imperfect observations or function mappings (more discussion later).

### 3.1.2 Provably Efficient Transfer in Q-Learning

A central objective in transfer learning is to establish theoretical guarantees that knowledge transfer leads to improved sample efficiency (Agarwal et al., 2023; Tirinzoni et al., 2020; Mann & Choe, 2013). However, these results do not apply readily to reward adaptation, which has a more restrictive problem setting that allows the analysis to be more targeted. In particular, in our approach, the efficiency of sample complexity lies in action pruning. Using Theorem 7 from Qu & Wierman (2020), the sample complexity bound is given by $\mathcal{O}\left(\frac{(|\mathcal{S}||\mathcal{A}|)^2 \, t_{\mathrm{mix}}}{(1-\gamma)^5 \, \varepsilon^2}\right)$, where $|\mathcal{S}|$ denotes the cardinality of the state space, $|\mathcal{A}|$ denotes the cardinality of the action space, $t_{\mathrm{mix}}$ is the mixing time of the underlying Markov chain induced by the policy, $\gamma \in (0,1)$ is the discount factor, and $\varepsilon > 0$ is the accuracy parameter. Q-M pruning strategy eliminates suboptimal actions while ensuring that the optimal action is preserved, thereby yielding a reduced action set $\tilde{A} \leq \mathcal{A}$. Furthermore, pruning may render some states of the original MDP unreachable, since transitions associated with pruned actions are removed. This is equivalent to cutting down the state space since the theory applies to space that is ergodic. Hence, action pruning directly impacts the $(|\mathcal{S}||\mathcal{A}|)^2$ term in the bound. While it is not straightforward to theoretically establish changes to the mixing time of the MDP after pruning, as long as the simplified transition structure leads to a mixing time that is not larger, sample efficiency will improve. Together, these effects would likely yield a *polynomial* reduction in sample complexity, as compared to the original MDP. It is also worth noting that action pruning not only benefits sample complexity but also regret. Even though these are separate concepts in RL, they are often related: the theoretical regret bounds often depend polynomially on the cardinality of the state and action space (Zhang et al. (2020); Bai et al. (2019)).

### 3.2 Q-Manipulation

In transfer learning, the source domains can pass information to the target domain to facilitate its learning. In RA such as Barreto et al. (2017), for example, what is passed includes 1) successor features, which are essentially discounted feature counts, and 2) source weights. The successor features of a source policy allow it to be evaluated easily under any new task given its weights. In Q-M, we assume the following to be passed: a) source reward function, and b) a lite-model of the environment (denoted by $\hat{T}(\cdot|s,a)$), where the lite-model captures neighboring information only. For example, the neighbors of a state after executing an action may be represented as a small region around that state. Since the lite-model does not model the distribution of such neighbors, learning it accurately only requires each transition to be visited at least once jointly while training the source behaviors. While it is still a strong assumption, it simplifies such joint learning. Optimality of Q-M is no longer guaranteed when this assumption fails. For the target domain, feature weights are assumed in Barreto et al. (2017) and $f$ is assumed in Q-M. Both may also be learned. From this perspective, Q-M is comparable to Barreto et al. (2017) in terms of resource demands.

In Q-M, we iteratively refine an upper and lower bound (UB and LB) of $Q_{\mathcal{R}}^*$ . These steps are formalized below:

**Definition 3.3** (Q-M Bellman Operators)**.** The Bellman operators for UB and LB in Q-M are mappings $\mathcal{T} : \mathbb{R}^{|S \times A|} \to \mathbb{R}^{|S \times A|}$ that satisfy, respectively:

$$(\mathcal{T}_{max}Q_k^{UB})(s,a) = \max_{s' \in \hat{T}(\cdot|s,a)} \left[ \mathcal{R}(s,a,s') + \gamma \max_{a'} Q_k^{UB}(s',a') \right] \tag{4}$$

$$(\mathcal{T}_{min}Q_k^{LB})(s,a) = \min_{s' \in \hat{T}(\cdot|s,a)} \left[ \mathcal{R}(s,a,s') + \gamma \max_{a'} Q_k^{LB}(s',a') \right] \tag{5}$$

More specifically, $\hat{T}(\cdot|s,a)$ denotes **1-step reachable states** from $s,a$. When $\hat{T}(\cdot|s,a)$ is given, it can be provided directly to the target domain. In practice, $\hat{T}(\cdot|s,a)$ can be estimated via memorization while learning the source behaviors. Such information can then be consolidated by the target domain. Optimality is no longer guaranteed when such an assumption does not hold. A similar practice can be adopted for learning the source reward functions when they are not available, which are combined via $f$ to compute the target reward function used above (i.e., $\mathcal{R}$). When $R_i$ is subject to noise, it may be approximated by the mean reward measured during source training. However, this approximation would also result in the loss of the optimality guarantee (see Section B.3). With deterministic domains, the Q-M Bellman operators above are exactly the operator in value iteration.

Note about lite-models: Zhuo & Kambhampati (2017) has explored the use of lite-models for planning. Lite models are incomplete or approximate models of a domain: they provide limited or coarse-grained information about the domain dynamics. While the lite-model considered in our work cannot be used directly for planning, it can support reinforcement learning, opening up a new possibility for transfer learning.

Similar to value iteration, the UB and LB can be initialized arbitrarily, established by the following theorem.

**Theorem 3.4** (Q-M Convergence)**.** $\mathcal{T} : \mathbb{R}^{|S \times A|} \to \mathbb{R}^{|S \times A|}$ *is a strict contraction such that the Q function converges to a unique fixed point for UB and LB, respectively, or more formally:*

$$\|\mathcal{T}'Q_A - \mathcal{T}'Q_B\|_\infty \leq \gamma \|Q_A - Q_B\|_\infty, \forall Q_A, Q_B \in \mathbb{R}^{|S \times A|}$$

$\|f\|_\infty = \sup_x |f(x)|$ and hence $\|Q_A - Q_B\|_\infty$ returns the maximum absolute difference between $Q_A(s,a)$ and $Q_B(s,a)$ under any $s,a$ above. Given that the Q values for UB and LB may be initialized arbitrarily, Next, we demonstrate that the bounds are valid upon convergence. This is established by the following theorem:

**Theorem 3.5.** *Given the standard Bellman operator $T$, $\mathcal{T}_{min}$, and $\mathcal{T}_{max}$ (discussed above), if the Q-functions are such that $Q_0^{LB} \leq Q_0 \leq Q_0^{UB}$, then*

$$Q_k^{LB} \leq Q_k \leq Q_k^{UB}$$

*holds for all $k \geq 0$. Thus, the fixed points $Q^*$, $Q^{LB}$, and $Q^{UB}$ produced by the respective operators, satisfy the same ordering in the limit as $k \to \infty$.*

### 3.2.1 Monotonic Q-Manipulation (M-Q-M)

We have shown that Q-M iteration process will converge to a fixed UB and LB, respectively. However, the updates are conservative in the sense that only the best or worst transitions are considered. This can lead to loose bounds even when we start with tighter bounds as initialization, which is counterproductive. Next, we consider new update rules to avoid the issue when we have a valid UB and LB to start with.

First, we formalize the new updates as follows:

**Upper Bound (UB)**

$$Q_0^{UB}(s,a) > Q^* \qquad \text{[Initialization]} \tag{6}$$

$$Q_{k+1}^{UB}(s,a) = \min \left( Q_k^{UB}(s,a), \max_{s' \in \hat{T}(\cdot|s,a)} \left[ \mathcal{R}(s,a,s') + \gamma \max_{a'} Q_k^{UB}(s',a') \right] \right) \tag{7}$$

**Lower Bound (LB)**

$$Q_0^{LB} < Q^* \qquad \qquad \text{[Initialization]} \tag{8}$$

$$Q_{k+1}^{LB}(s,a) = \max\left(Q_k^{LB}(s,a), \min_{s'\in\hat{T}(\cdot|s,a)}\left[\mathcal{R}(s,a,s') + \gamma\max_{a'}Q_k^{LB}(s',a')\right]\right) \tag{9}$$

This update guarantees that the upper and lower bounds are monotonically non-increasing (for the upper bound) and non-decreasing (for the lower bound). At every iteration, the bounds satisfy the condition: $Q^{LB} \leq Q^* \leq Q^{UB}$. The outermost max/min ensures $Q^{UB} \geq Q^* \geq Q^{LB}$ throughout the iterative processes via simple induction. When the source reward functions are noisy, it requires their min/ max noise to be used in the updates. Similar to Q-M, when $R_i$ is noisy, using mean reward may lead to inaccuracy due to approximate $R_i$ as well as inaccurate source Q functions derived from $R_i$. In such cases the soundness of the bounds cannot be guaranteed (refer to Section B.3). Next, before discussing the initializations, we show that such processes converge to a fixed point in M-Q-M.

**Definition 3.6.** (M-Q-M Bellman Operators) The min and max Bellman operator for UB and LB in M-Q-M are mappings $\mathcal{T} : \mathbb{R}^{|S\times A|} \to \mathbb{R}^{|S\times A|}$ that satisfy, respectively:

$$(\mathcal{T}_{min}Q_k^{UB})(s,a) = \min\left(Q_k^{UB}(s,a), \max_{s'\in\hat{T}(\cdot|s,a)}\left[\mathcal{R}(s,a,s') + \gamma\max_{a'}Q_k^{UB}(s',a')\right]\right)$$

$$(\mathcal{T}_{max}Q_k^{LB})(s,a) = \max\left(Q_k^{LB}(s,a), \min_{s'\in\hat{T}(\cdot|s,a)}\left[\mathcal{R}(s,a,s') + \gamma\max_{a'}Q_k^{LB}(s',a')\right]\right)$$

Since the theoretical results for the min and max operators are similar, we do not distinguish between them below but provide separate proofs for them in the appendix.

**Theorem 3.7** (M-Q-M Convergence)**.** *The iteration process introduced by the Bellman operator in M-Q-M satisfies*

$$\|\mathcal{T}Q_k - \mathcal{T}Q_{k+1}\|_\infty \leq \gamma\|Q_k - Q_{k+1}\|_\infty, \forall Q_k, Q_{k+1} \quad \in \mathbb{R}^{|S\times A|}$$

*under the initialization assumption of Equation* (6) *and Equation* (8) *such that the Q function converges to a fixed point.*

This process converges to a fixed point, since the difference between two consecutive iterations always decreases. However, it turns out that the fixed point may not necessarily be unique, as with value iteration.

**Theorem 3.8.** *The Bellman operator in M-Q-M specifies only a non-strict contraction in general:*

$$\left\|\mathcal{T}Q - \mathcal{T}\widehat{Q}\right\|_\infty \leq \left\|Q - \widehat{Q}\right\|_\infty$$

This result is interesting since it identifies another case where non-strict contraction results in a fixed point other than the identity map.

**Corollary 3.9** (Non-uniqueness)**.** *The fixed point of the iteration process in M-Q-M may not be unique.*

In our evaluation, we observe that the fixed point found by the M-Q-M iteration process depends on the initialization. Another observation is that the Bellman operator in M-Q-M appears almost identical to that in value iteration when the MDP is deterministic. In such cases, we observe that Q-M and M-Q-M often results in zero-shot learning when the upper and lower bounds converge to $Q_\mathcal{R}^*$.

### 3.2.2 Initializing the Bounds

In order to use M-Q-M updates, the user must provide some correct bounds to start with. To relax such a requirement, next, we show how high quality initialization can be automatically computed for restrictive sets of problems. Computing such an initialization requires additional knowledge to be transferred from the source to the target domains, which can be done by transferring the source domain $Q_i^*$ and "worst return" policy $Q_i^\mu$ values to the target domain.

A simple way to initialize the bounds would be to identify the most positive and negative rewards and compute the sums of their geometric sequences via the discount factor, respectively. However, these bounds

are likely to be too conservative to be useful since the iteration processes may converge undesirably due to non-unique fixed points. Intuitively, we would like the bounds to be tight initially to yield the best results. However, computing bounds for the target behavior based on information from the source behaviors only is not a trivial task. Next, we show situations where additional assumptions hold such that we can provide more desirable initializations. In particular, we will show next how different forms of the combination function $f$ in Eq. 3 can affect the initializations.

**Linear Combination Function:** First, we consider the case when the target reward function is a linear function of the source reward functions. In such cases, if the agent maintains both $Q_i^\mu$'s and $Q_i^*$'s while learning the source behaviors, we propose the initializations as follows. Note that $Q_i^\mu$ can be obtained conveniently while learning the source behaviors based on Lemma 3.1.

**Lemma 3.10.** *When $\mathcal{R} = \sum_{i=1}^n c_i R_i$ with $c_i \geq 0$, the upper and lower bounds of $Q_\mathcal{R}^*$ are, respectively, $Q_0^{UB} = \sum_{i=1}^n c_i Q_i^*$ and $Q_0^{LB} = \max_i \left[ c_i Q_i^* + \sum_j c_j Q_j^\mu \right]$, where $j \in \{1 : n\} \setminus i$.*

**Nonlinear Combination Function:** Handling a nonlinear combination is more complicated and deriving tight bounds that are guaranteed to be correct is difficult. Instead, we propose approximate bounds for a monotonically increasing and positive function $f$ as follows: $Q_0^{UB} = f(Q_{|R_1|}^*, Q_{|R_2|}^*, \ldots, Q_{|R_n|}^*)$ and $Q_0^{LB} = -f(Q_{|R_1|}^*, Q_{|R_2|}^*, \ldots, Q_{|R_n|}^*)$.

Using the bounds above requires the agent to maintain $Q_{|R_i|}^*$'s (representing Q function with respect to reward $|R_i|$). Since these bounds are approximate, they do not guarantee correctness for M-Q-M in general, meaning that actions belonging to the optimal policy may be pruned. However, we show that they work well in practice in our evaluation.

### 3.2.3 Noisy Combination Function

When the combination function is not known exactly but can be modeled with an additional noise component, such that $\mathcal{R} = f(R_1 \ldots R_n) + N$, and we know the range of the noise (i.e., $N_{min}$ and $N_{max}$). We can consider such situations by augmenting the $\mathcal{R}(s, a, s')$ in Eqs. 7 and 9 with $N_{max}$ and $N_{min}$, respectively. We must also update the initialization of the bounds using $Q^{UB} = Q^{UB} + N_{max} \times \frac{1 - \gamma^{t_{max}}}{1 - \gamma}$ and $Q^{LB} = Q^{LB} + N_{min} \times \frac{1 - \gamma^{t_{max}}}{1 - \gamma}$, where $t_{max}$ is the maximum steps in an episode. Note however that such modifications will likely reduce the efficacy of Q-M.

### 3.3 Action Pruning in Q-M:

If an action $a$'s lower bound is higher than some other action $\hat{a}$'s upper bound under a state $s$, then $\hat{a}$ can be pruned for that state. This allows us to reduce the action space per each different state, which contributes to faster convergence (refer to Sec. 3.1.2). For empirical purposes and to avoid numerical instability, we use a threshold ($\Delta$) and prune only if $Q^{LB}(s, a) - Q^{UB}(s, \hat{a}) \geq \Delta$. When the source domain's Q values are computed using value iteration with a stopping threshold $\epsilon$, $\Delta$ can be set to be $2\epsilon \frac{\gamma}{1 - \gamma}$ to ensure that no actions would be wrongly pruned. When the Q values are approximated (such as via Q learning) and $\epsilon$ is unknown, setting $\Delta$ would not be so straightforward and we delay its treatment to future work. When the upper and lower bounds are sound, the optimal policies are preserved.

**Theorem 3.11.** *[Optimality] For reward adaptation with Q variants, the optimal policies in the target domain remain invariant under Q-M and M-Q-M (under their ideal settings discussed, respectively) when the upper and lower bounds are initialized correctly.*

## 4 Evaluation

### 4.1 Baselines

The primary objective here is to evaluate the performance of Q-M using target time to threshold Taylor & Stone (2009) and analyze its benefits and limitations. We compare Q-M against three methods: SFQL

Barreto et al. (2017), a state-of-the-art approach for reward adaptation; SQB Adamczyk et al. (2024), which clips the Bellman error using a prior $Q$-function to accelerate learning; and standard Q-Learning (QL) without any knowledge transfer as a baseline. To initialize learning for SFQL, we evaluate the given source behaviors in the target domain to compute a bootstrap Q-function as described in the generalized policy improvement theorem in Barreto et al. (2017). Additional results for the running time taken by the Q-M iteration process are reported in Sec. A.4.2. We keep the hyperparameters for Q-Learning the same across the different methods (refer to Sec. A.4.1).

## 4.2 Evaluation Design

Since we are interested in demonstrating Q-M (short for Q-M/M-Q-M unless separately noted) as a more robust knowledge transfer method than SFQL or SQB, we design the evaluation domains such that the target behaviors are substantially different from the source behaviors in most of them (similar to the situation in Dollar-Euro). Designing evaluations this way also provides an opportunity to study negative transfer in transfer learning. Details on how the source and target behaviors are designed are in the appendix. For SFQL, policy evaluation of the source behaviors, required to bootstrap target learning, is achieved via value iteration on the target. To analyze the theoretical properties of Q-M, we assume access to accurate lite-models, reward functions of the source behaviors, and Q-variants (only for M-Q-M and computed using value iteration). In the appendix, we use memorization to estimate these from source domains and these variants have comparable performance to their exact counterparts. For M-Q-M, we use the initializations described in Sec. 3.2.2.

One observation about Q-M is that the computation of UB and LB is affected substantially by the stochastic branching factor (SBF) of a domain, as evident in Eqs. 4, 5, 7 and 9. SBF here is defined as the maximum number of next states reachable (or with a nonzero transition probability) from any state and action pair Intuitively, the less stochastic the domain is, the more the Bellman updates in Q-M resemble that in value iteration: Q-M updates in deterministic domains are exactly value iteration updates, resulting in zero-shot learning. To demonstrate the influence of SBF, for each evaluation domain, we test with a range of SBF values. At the same time, the number of reachable states from a given state is allowed to vary and is randomly chosen between 1 and a set SBF. We first evaluate with gridworld domains where target R is a linear combination of source rewards. We also visualize actions pruning in a chosen domain to illustrate its operation. To evaluate the generality, evaluations are further conducted with autogenerated MDPs and with linear and non-linear combination functions Finally, we study the effectiveness of Q-M under noisy combination functions, which analyzes the situations when the combination functions must be learned but noise can be bounded.

All evaluations are averaged over 30 runs. In the convergence plots, we indicate the mean with a solid line, and the shaded region represents a 95% confidence interval. More details about the evaluation settings, along with a detailed description of all the domains, including the design of source and target behaviors, are reported in the appendix.

## 4.3 Gridworld and Linear Combination Function

In this evaluation, we compared Q-M and M-Q-M with the baselines in simulation domains that include Racetrack, Dollar-Euro, and Frozen Lake domain with linear combination functions. The convergence plots are shown in Fig. 2. In each subfigure, we show the SBF used (labeled at the top). We observe that M-Q-M converges substantially faster than the baselines in all three domains. Depending on how many actions are pruned under traditional Q-M, its performance lies between M-Q-M and QL. In Sec. 4.3.1, we show how action pruning differs between M-Q-M and Q-M, which results in such an effect on convergence. However, as expected, the performance of Q-M and M-Q-M are negatively impacted as SBF increases. An interesting observation is the performance of SFQL. SFQL seems to struggle with these domains, especially racetrack and frozen lake domains. Since the source behaviors differ much from the target behavior, knowledge transfer in SFQL based on combining the source behaviors can actually misguide the learning process. It is worth mentioning that SFQL eventually converged to the optimal policy after we allowed it to train with more episodes. In addition, we also observe that Q-M in deterministic scenarios (leftmost subfigures when SBF = 1) result in zero-shot learning: their iterative processes for computing UB and LB both converge to $Q^*_{\mathcal{R}}$. Similar to SFQL, SQB also struggles in the racetrack and frozen lake domain. This result demonstrates that

Q-M is indeed more robust against negative transfer, and thus represents a more general knowledge transfer method that does not depend on the similarity between the source and target behaviors.

Figure 2: Convergence plots for Dollar Euro (top), Racetrack (mid), and Frozen Lake (bottom).

### 4.3.1 Analysis of Action Pruning

For gridworld domains (with 4 actions), to understand the states where actions are pruned, we plot heat-maps (refer Fig. 3 for the Dollar Euro domain). In all three domains, we observe significant pruning around the terminal states. In addition, we also observe that fewer actions are pruned as SBF increases. The following color codes are used: initial state = yellow and goal states = green. We use different shades of blue to illustrate how many actions are pruned in a state: the lighter the color, the fewer the actions remain. Upon comparing Dollar-Euro domain's action pruning using M-Q-M and Q-M, we observe that Q-M results in pruning fewer actions (as shown in the Fig. 3). The additional information used by M-Q-M is able to anchor the values to better bounds than the unique fixed point identified by Q-M, which results in more pruning opportunities in M-Q-M. As the SBF increases, Q-M prunes out fewer actions, and so performance becomes similar to QL. This trend is consistent across other domains as well.

### 4.4 Autogenerated MDP with Linear and Non Linear Combination Function

In order to test generalization beyond gridworld and linear combination function, we evaluated with auto-generated MDP where $T$ is randomly generated in each run. The terminal states were held fixed as well as their terminal rewards. The convergence plots are presented in Fig. 4, averaged over 30 different MDPs. Similarly, we can observe that M-Q-M performs the best in both domains. We also observe that when the

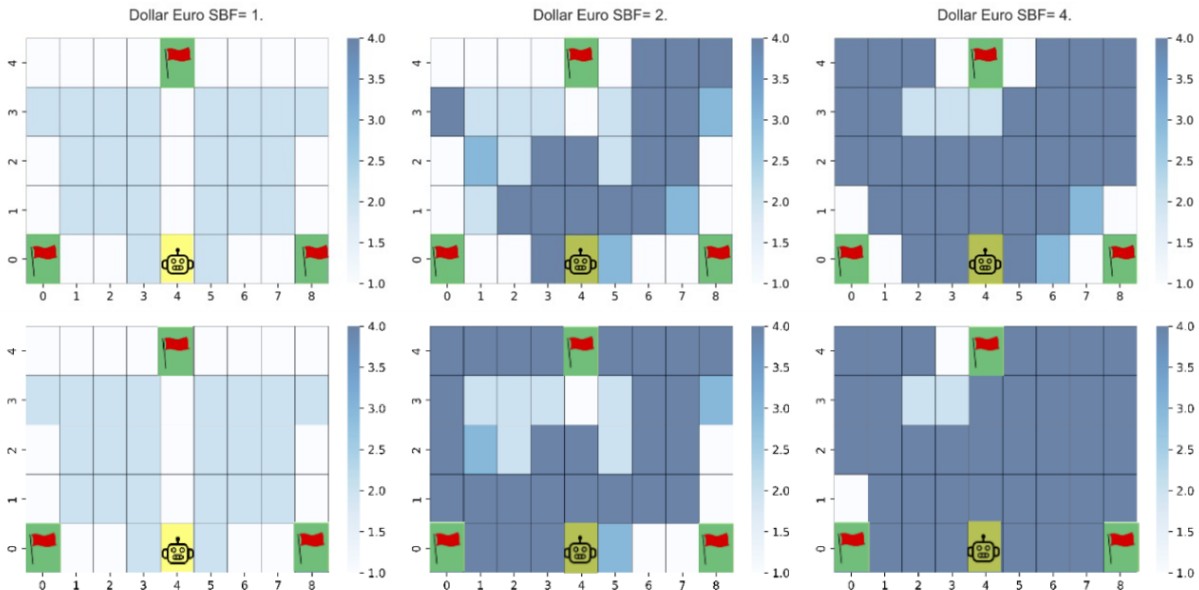

Figure 3: Heat-maps illustrating action pruning in the Dollar Euro domain using M-Q-M (top) and Q-M (bottom). Lighter shade of blue indicates fewer action remain after pruning.

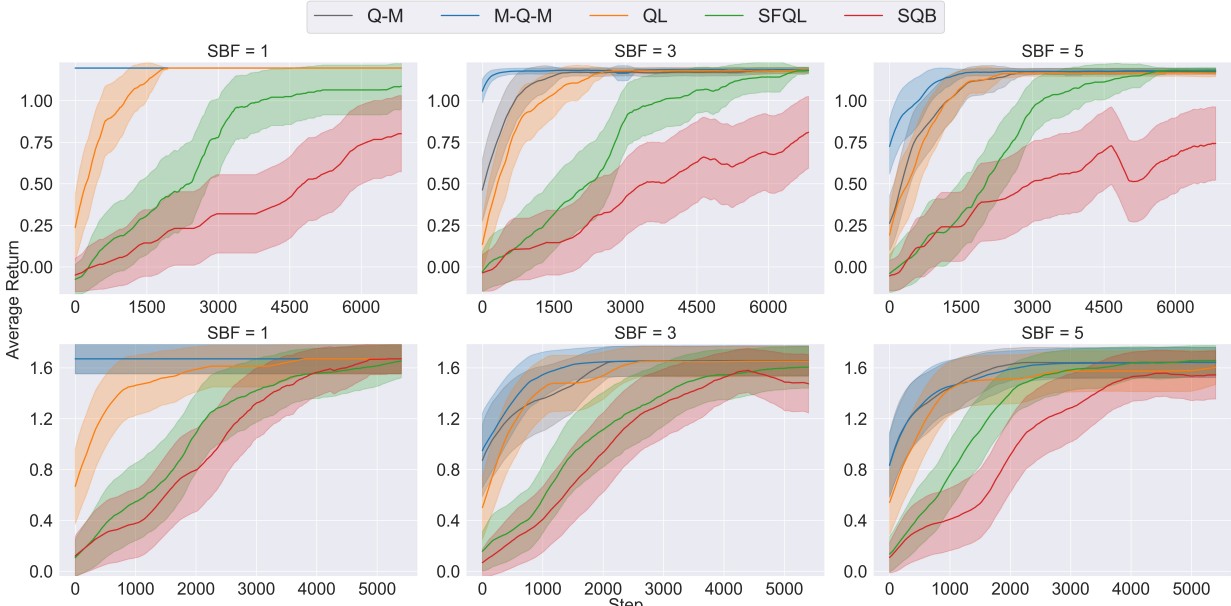

Figure 4: Convergence plots for auto-generated domains: $\mathcal{R} = R_1 + R_2$ (top) and $\mathcal{R} = (R_1 + R_2)^3$ (bottom).

combination function is a nonlinear combination, M-Q-M and Q-M's performance drops due to reduced action pruning. It demonstrates that our methods can generalize to MDPs beyond gridworld dynamics and different target reward combinations.

### 4.5 Noisy Combination Function

We aim to evaluate how Q-M would perform under noisy combination functions and how noise affects its performance. We used the same setting as the autogenerated MDP described above. We consider a situation where the combination function is not exactly known but can be modeled by using a noise component: $\mathcal{R} = R_1 + R_2 + N$. Assuming the knowledge of $N_{\min}$ and $N_{\max}$, we updated the initializations and Bellman

updates for M-Q-M. The convergence plots are presented in Fig. 5, demonstrating the diminishing boost in performance as the number of actions pruned decreases with increase in the magnitude of noise. As expected, we observe that noise has an impact on the efficacy of M-Q-M: the more noise, the smaller the performance gain. It is important to note that the maximum magnitude of noise that allows action pruning depends on MDP reward design. However, it is promising to observe that M-Q-M can still be effective under such noisy situations since it can greatly expand the applicability of M-Q-M. For instance, when the functional relationship is unknown, we can apply regression to fit the source reward functions to the observed target rewards under an assumed functional form based on domain expertise; noise can be incorporated to handle regression error.

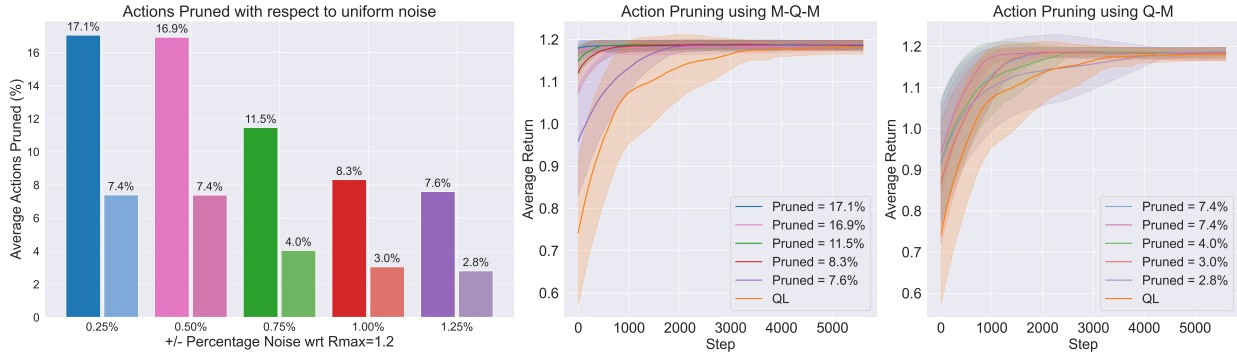

Figure 5: Performance under varying noise in auto-generated domains. Left: Actions pruned (%) vs. noise. Middle: M-Q-M convergence. Right: Q-M convergence. (color: dark = M-Q-M, lighter = Q-M).

## 5 Limitations and Future Work

**Convergence Quality for M-Q-M**: The convergence behavior of M-Q-M depends critically on its initialization relative to the convergence point of Q-M. In the case where the initialization of M-Q-M strictly dominates the fixed point of Q-M (i.e., higher for upper bounds and lower for lower bounds,$\forall s, a$), M-Q-M would perform at least as well as Q-M. However, the situation is less clear when the quality of the initialization is less known. This raises an open question regarding the quality of convergence for M-Q-M: Under what conditions on the initialization does M-Q-M yield strictly tighter bounds or improved performance compared to Q-M? Formal characterization of such conditions, potentially in terms of partial dominance or monotonicity properties over subsets of the state-action space, could offer deeper theoretical insights and guide more effective initialization strategies. Exploring this direction represents a compelling avenue for future work.

**Safety Considerations:** Safety is critical when applying Q-M in real-world settings. For instance, in autonomous driving, $Q_i^*$ may prioritize avoiding obstacles, ensuring safe navigation. In contrast, learning $Q_i^\mu$ could result in reckless behavior, like colliding with obstacles, which is unsafe and undesirable. Therefore, selecting a safe, ethical behavior for learning Q-variants becomes essential when designing Q-M systems for real-world applications. One possible solution is to leverage safe RL methods, such as shielding Alshiekh et al. (2018), to ensure safety while learning Q-variants.

**Scaling to Real-World Domains:** While effective in discrete settings, Q-M must overcome challenges to scale to continuous spaces, where Q-value bounds are harder to initialize and refine. This raises a central challenge: how to effectively initialize and refine Q-value bounds when the space is uncountably infinite. One solution lies in leveraging function approximation to estimate these bounds efficiently. Additionally, identifying one-step reachable neighbors becomes more complex in continuous spaces. A potential approach is to approximate these neighbors via uniform sampling within a bounded radius in the state space, or to derive closed-form functions that produce finite, representative next states. Approximation errors may cause Q-M to prune optimal actions, degrading performance. To ensure reliability, pruning should be used cautiously and combined with techniques that mitigate approximation bias. For example, setting $\Delta$ (refer to Sec. 3.3) should be done empirically due to the fact that the error in the estimated value function is unknown.

**Generalization and Domain Adaptation:** Q-M currently assumes a known relationship between source and target rewards, limiting its use when target rewards are unknown. Extending Q-M to learn or infer these relationships would improve adaptability. Its pruning strategy could also support dynamics transfer (off-dynamics RL), expanding applications to areas like sim2real transfer.

Q-M provides a promising foundation for transfer in RL. Realizing its practical impact will require addressing safety, scalability, approximation errors, and generalization, each offering challenges and opportunities for broader real-world adoption.

## 6 Conclusions

In this paper, we studied reward adaptation, the problem where the learning agent adapted to a target reward function based on the existing source behaviors under the same MDP except for $R$. We propose 2 methods 1) Q-Manipulation (Q-M) and its extension, 2) Monotonic Q-Manipulation (M-Q-M) as novel, theoretically grounded approaches for reward adaptation in reinforcement learning. By leveraging source Q-function variants, these methods compute tight bounds on the target Q-function to safely prune suboptimal actions. We formally proved that our approach converged and retained optimality under correct initializations. Empirically, we showed that Q-M and M-Q-M were substantially more efficient than the baselines in domains where the source and target behaviors differ, and generalizable under different randomizations. We also applied Q-M to noisy combination functions to extend its applicability. As such, our methods offer a robust framework for leveraging prior knowledge in reinforcement learning, advancing the state of transfer and continual learning. Our work also opens up many future opportunities, such as addressing continuous state and action spaces and handling different domain dynamics (in addition to reward functions) as in domain adaptation.

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

# A Appendix

## A.1 Theoretical Proofs

**Lemma 3.1**

$$Q_R^\mu(s,a) = \min_\pi \left[ \mathbb{E}\left[ \sum_{t=0}^\infty \gamma^t r_t | s_0, \pi \right] \right]$$
$$= -\max_\pi \left[ \mathbb{E}\left[ \sum_{t=0}^\infty -\gamma^t r_t | s_0, \pi \right] \right] \tag{10}$$
$$= -Q_{-R}^*(s,a)$$

**Lemma 3.10** When $\mathcal{R} = \sum c_i R_i$ where $c_i \geq 0$ , an upper and lower bound of $Q_\mathcal{R}^*$ are given, respectively, by:

$$Q_0^{UB} = \sum_{i=1}^n c_i Q_i^*$$
$$Q_0^{LB} = \max_i [c_i Q_i^* + \sum_j c_j Q_j^\mu] \text{ where } j \in \{1 : n\} \setminus i \tag{11}$$

*Proof.* From definition, we have:

$$c_i Q_i^\pi = \max_\pi \left[ \mathbb{E}\left[ c_i r_{i,0} + \gamma c_i r_{i,1} + \ldots + \gamma^n c_i r_{i,n} | s_0, \pi \right] \right] \tag{12}$$

By reorganizing the reward components, we have:

$$\sum_i c_i Q_i^\pi = Q_{\sum_i c_i R_i}^\pi \tag{13}$$

Denote the optimal policy under the target reward function $\mathcal{R}$ as $\pi^*$, given $c_i \geq 0$, we can derive that

$$\sum_i c_i Q_i^* \geq \sum_i c_i Q_i^{\pi^*} = Q_\mathcal{R}^* \tag{14}$$

For the lower bound, we have:

$$\max_i(c_i Q_i^* + \sum_{j \neq i} c_j Q_j^\mu) \leq c_k Q_k^* + \sum_{j \neq k} c_j Q_j^{\pi_k^*}$$

where $k$ denotes the best choice of $i$ from the left

$$\leq \max_\pi(c_i Q_i^\pi + \sum_{j \neq i} c_j Q_j^\pi) \tag{15}$$

$$= Q_\mathcal{R}^*$$

$\square$

Next, we present a few lemmas that are used in the proof of our theorems:

**Lemma A.1.**

$$\left| \max_a f(a) - \max_a g(a) \right| \leq \max_a |f(a) - g(a)|.$$

*Proof.* Assume without loss of generality that $\max_a f(a) \geq \max_a g(a)$, and denote $a^* = \arg\max_a f(a)$. Then,

$$\left| \max_a f(a) - \max_a g(a) \right| = \max_a f(a) - \max_a g(a) = f(a^*) - \max_a g(a) \leq f(a^*) - g(a^*) \leq \max_a |f(a) - g(a)|.$$

This concludes the proof. $\square$

**Lemma A.2.**
$$\left| \min_a f(a) - \min_a g(a) \right| \leq \max_a |f(a) - g(a)|.$$

*Proof.* Assume without loss of generality that $f(a^*) = \min_a f(a) \geq \min_a g(a) = g(b^*)$. Then,

$$\max_a |f(a) - g(a)| \geq |f(b^*) - g(b^*)| \geq f(b^*) - g(b^*) \geq f(a^*) - g(b^*) = \left| \min_a f(a) - \min_a g(a) \right|$$

This concludes the proof. $\qquad\square$

**Theorem 3.4** [Q-M Convergence] $\mathcal{T} : \mathbb{R}^{|S \times A|} \to \mathbb{R}^{|S \times A|}$ is a strict contraction such that the $Q$ function converges to a unique fixed point for UB and LB, respectively, or more formally:

$$\|\mathcal{T}'Q_A - \mathcal{T}'Q_B\|_\infty \leq \gamma \|Q_A - Q_B\|_\infty, \forall Q_A, Q_B \quad \in \mathbb{R}^{|S \times A|}$$

*Proof.* For any two Q-functions, denoted $Q_A^{LB}$ and $Q_B^{LB}$, the distance between their transformations under the operator is strictly smaller than their original distance, scaled by $\gamma$. We demonstrate this for $\mathcal{T}_{min}$. The proof for $\mathcal{T}_{max}$ follows a symmetric argument.

We want to show:
$$\|\mathcal{T}_{min}Q_A^{LB} - \mathcal{T}_{min}Q_B^{LB}\|_\infty \leq \gamma \|Q_A^{LB} - Q_B^{LB}\|_\infty.$$

Consider the absolute difference for an arbitrary state-action pair $(s, a)$:

$$\left| (\mathcal{T}_{min}Q_A^{LB})(s, a) - (\mathcal{T}_{min}Q_B^{LB})(s, a) \right|$$

$$= \left| \min_{s' \in \hat{T}(\cdot|s,a)} \left[ \mathcal{R}(s, a, s') + \gamma \max_{a'} Q_A^{LB}(s', a') \right] - \min_{s' \in \hat{T}(\cdot|s,a)} \left[ \mathcal{R}(s, a, s') + \gamma \max_{a'} Q_B^{LB}(s', a') \right] \right|$$

$$\leq \max_{s' \in \hat{T}(\cdot|s,a)} \left| \left[ \mathcal{R}(s, a, s') + \gamma \max_{a'} Q_A^{LB}(s', a') \right] - \left[ \mathcal{R}(s, a, s') + \gamma \max_{a'} Q_B^{LB}(s', a') \right] \right| \quad \text{(Lemma A.2)}$$

$$= \max_{s' \in \hat{T}(\cdot|s,a)} \left| \gamma \max_{a'} Q_A^{LB}(s', a') - \gamma \max_{a'} Q_B^{LB}(s', a') \right|$$

$$= \gamma \max_{s' \in \hat{T}(\cdot|s,a)} \left| \max_{a'} Q_A^{LB}(s', a') - \max_{a'} Q_B^{LB}(s', a') \right|$$

$$\leq \gamma \max_{s' \in \hat{T}(\cdot|s,a)} \max_{a'} \left| Q_A^{LB}(s', a') - Q_B^{LB}(s', a') \right| \quad \text{(Lemma A.1)}$$

$$\leq \gamma \|Q_A^{LB} - Q_B^{LB}\|_\infty$$

This inequality holds for all $(s, a)$, so taking the maximum over all state-action pairs gives:

$$\|\mathcal{T}_{min}Q_A^{LB} - \mathcal{T}_{min}Q_B^{LB}\|_\infty \leq \gamma \|Q_A^{LB} - Q_B^{LB}\|_\infty.$$

Since $\gamma \in [0, 1)$, the operator $\mathcal{T}_{min}$ is a contraction. By the Banach Fixed-Point Theorem, this implies that $\mathcal{T}_{min}$ has a unique fixed point in the space of bounded Q-functions. $\qquad\square$

**Theorem 3.5** Given the standard Bellman operator $T$, $\mathcal{T}_{min}$, and $\mathcal{T}_{max}$ (discussed above), if the Q-functions are such that $Q_0^{LB} \leq Q_0 \leq Q_0^{UB}$, then
$$Q_k^{LB} \leq Q_k \leq Q_k^{UB}$$
holds for all $k \geq 0$. Thus, the fixed points $Q^*$, $Q^{LB}$, and $Q^{UB}$ produced by the respective operators, satisfy the same ordering in the limit as $k \to \infty$.

*Proof.* First, we establish two key properties of the operators.

**Lemma A.3** (Operator Properties)**.**

1. ***Monotonicity:*** *For any two Q-functions $Q_A, Q_B$ such that $Q_A \leq Q_B$ (pointwise), we have $T_\Omega Q_A \leq T_\Omega Q_B$ for any of the three operators $\Omega \in \{standard, LB, UB\}$. This is because $\max_{a'} Q_A \leq \max_{a'} Q_B$, and the* min*,* max*, and expectation operators all preserve this inequality. (for formal proof, refer to proposition 3.6 in Kadurha et al. (2025))*

2. ***Ordering:*** *For any Q-function $Q$, we have $\mathcal{T}_{min}Q \leq TQ \leq \mathcal{T}_{max}Q$. This follows from the definition of expectation that the minimum of a set of values is less than or equal to their expectation, which is less than or equal to their maximum.*

To prove $Q^{LB} \leq Q^* \leq Q^{UB}$ We proceed by mathematical induction:

**Base Case (k=0):**
$$Q_0^{LB} \leq Q_0 \leq Q_0^{UB} \quad \text{(given)}$$

**Inductive Hypothesis**: Assume that for some $k \geq 0$, we have the inductive hypothesis:
$$Q_k^{LB} \leq Q_k \leq Q_k^{UB}$$

**Inductive Step**: First, we prove the left-hand inequality, $Q_{k+1}^{LB} \leq Q_{k+1}$:

$$
\begin{aligned}
Q_{k+1}^{LB} &= \mathcal{T}_{min}Q_k^{LB} && \text{(by definition)} \\
&\leq \mathcal{T}_{min}Q_k && \text{(by monotonicity and induction hypothesis)} \\
&\leq TQ_k && \text{(by operator ordering)} \\
&= Q_{k+1} && \text{(by definition)}
\end{aligned}
$$

Next, we prove the right-hand inequality, $Q_{k+1} \leq Q_{k+1}^{UB}$:

$$
\begin{aligned}
Q_{k+1} &= TQ_k && \text{(by definition)} \\
&\leq \mathcal{T}_{max}Q_k && \text{(by Operator Ordering)} \\
&\leq \mathcal{T}_{max}Q_k^{UB} && \text{(by Monotonicity and induction hypothesis)} \\
&= Q_{k+1}^{UB} && \text{(by definition)}
\end{aligned}
$$

Since both sides of inequalities hold, we have:
$$Q_{k+1}^{LB} \leq Q_{k+1} \leq Q_{k+1}^{UB}$$

Thus, by the principle of mathematical induction, the ordering
$$Q_k^{LB} \leq Q_k \leq Q_k^{UB}$$

holds for all $k \geq 0$. As $k \to \infty$, the inequality also holds for fixed points of individual operators. $\quad\square$

**Theorem 3.7** [Convergence] The iteration process introduced by the Bellman operator in M-Q-M satisfies
$$\|\mathcal{T}Q_k - \mathcal{T}Q_{k+1}\|_\infty \leq \gamma\|Q_k - Q_{k+1}\|_\infty, \forall Q_k, Q_{k+1} \in \mathbb{R}^{|S \times A|}.$$

such that the $Q$ function converges to a fixed point.

*Proof.* **1) Upper Bound**

The operator $\mathcal{T}_{\min}$ for the upper bound is defined as follows:

$$Q_{k+1}^{UB}(s,a) = (\mathcal{T}_{\min}Q_k^{UB})(s,a) = \min\left(Q_k^{UB}(s,a), \max_{s' \in \hat{T}(\cdot|s,a)}\left[\mathcal{R}(s,a,s') + \gamma \max_{a'} Q_k^{UB}(s',a')\right]\right) \quad (16)$$

where $\hat{T}(\cdot|s,a)$ denotes reachable states from $s,a$.

We consider the change of difference between $Q$ values between before and after the modified Bellman update (i.e., the difference between $\left|Q_k^{UB}(s,a) - Q_{k+1}^{UB}(s,a)\right|$ and $\left|Q_{k+1}^{UB}(s,a) - Q_{k+2}^{UB}(s,a)\right|$):

**Case 1:**   If the first elements were the smaller values for computing both $Q_{k+1}^{UB}$ and $Q_{k+2}^{UB}$ in Eq. 16:

$$Q_{k+1}^{UB}(s,a) = Q_k^{UB}(s,a)$$

$$Q_{k+2}^{UB}(s,a) = Q_{k+1}^{UB}(s,a)$$

$$\left|Q_{k+1}^{UB}(s,a) - Q_{k+2}^{UB}(s,a)\right| = |Q_k^{UB}(s,a) - Q_{k+1}^{UB}(s,a)| = 0$$

**Case 2:** If the second element in min was the smaller value for computing $Q_{k+1}^{UB}$ and the first element in min was the smaller value for $Q_{k+2}^{UB}$:

$$Q_{k+1}^{UB}(s,a) = \max_{s'\in\hat{T}(\cdot|s,a)}\left[\mathcal{R}(s,a,s') + \gamma\max_{a'}Q_k^{UB}(s',a')\right]$$

$$Q_{k+2}^{UB}(s,a) = Q_{k+1}^{UB}(s,a)$$

$$\left|Q_{k+1}^{UB}(s,a) - Q_{k+2}^{UB}(s,a)\right| = 0$$

**Case 3:** If the first element in min was the smaller value for computing $Q_{k+1}^{UB}$ and the second element in min was the smaller value for $Q_{k+2}^{UB}$:

$$Q_{k+1}^{UB}(s,a) = Q_k^{UB}(s,a) \leq \max_{s'\in\hat{T}(\cdot|s,a)}\left[\mathcal{R}(s,a,s') + \gamma\max_{a'}Q_k^{UB}(s',a')\right] \text{(Eq. 16)} \tag{17}$$

$$Q_{k+2}^{UB}(s,a) = \max_{s'\in\hat{T}(\cdot|s,a)}\left[\mathcal{R}(s,a,s') + \gamma\max_{a'}Q_{k+1}^{UB}(s',a')\right]$$

$$\left|Q_{k+1}^{UB}(s,a) - Q_{k+2}^{UB}(s,a)\right|$$

$$= Q_k^{UB}(s,a) - \max_{s'\in\hat{T}(\cdot|s,a)}\left[\mathcal{R}(s,a,s') + \gamma\max_{a'}Q_{k+1}^{UB}(s',a')\right]$$

$$\leq \max_{s'\in\hat{T}(\cdot|s,a)}\left[\mathcal{R}(s,a,s') + \gamma\max_{a'}Q_k^{UB}(s',a')\right] - \max_{s'\in\hat{T}(\cdot|s,a)}\left[\mathcal{R}(s,a,s') + \gamma\max_{a'}Q_{k+1}^{UB}(s',a')\right] \text{(Eq. 17)}$$

$$\leq \left|\max_{s'\in\hat{T}(\cdot|s,a)}\left[\mathcal{R}(s,a,s') + \gamma\max_{a'}Q_k^{UB}(s',a')\right] - \max_{s'\in\hat{T}(\cdot|s,a)}\left[\mathcal{R}(s,a,s') + \gamma\max_{a'}Q_{k+1}^{UB}(s',a')\right]\right|$$

$$\leq \gamma\max_{s'\in\hat{T}(\cdot|s,a)}\left|\max_{a'}Q_k^{UB}(s',a') - \max_{a'}Q_{k+1}^{UB}(s',a')\right| \text{ (Lemma A.1)}$$

$$\leq \gamma\max_{s'\in\hat{T}(\cdot|s,a)}\max_{a'}\left|Q_k^{UB}(s',a') - Q_{k+1}^{UB}(s',a')\right| \text{ (Lemma A.1)}$$

$$\leq \gamma\|Q_k^{UB}(s,a) - Q_{k+1}^{UB}(s,a)\|_\infty$$

**Case 4:**   If the second elements in min were the smaller values for both $Q_{k+1}^{UB}$ and $Q_{k+2}^{UB}$:

$$Q_{k+1}^{UB}(s,a) = \max_{s'\in\hat{T}(\cdot|s,a)}\left[\mathcal{R}(s,a,s') + \gamma\max_{a'}Q_k^{UB}(s',a')\right]$$

$$Q_{k+2}^{UB}(s,a) = \max_{s'\in\hat{T}(\cdot|s,a)}\left[\mathcal{R}(s,a,s') + \gamma\max_{a'}Q_{k+1}^{UB}(s',a')\right]$$

$$\left|Q_{k+1}^{UB}(s,a) - Q_{k+2}^{UB}(s,a)\right|$$

$$= \left|\max_{s'\in\hat{T}(\cdot|s,a)}\left[\mathcal{R}(s,a,s') + \gamma\max_{a'}Q_k^{UB}(s',a')\right] - \max_{s'\in\hat{T}(\cdot|s,a)}\left[\mathcal{R}(s,a,s') + \gamma\max_{a'}Q_{k+1}^{UB}(s',a')\right]\right|$$

$$\leq \gamma\|Q_k^{UB}(s,a) - Q_{k+1}^{UB}(s,a)\|_\infty \text{ (similar to Case 3 above)}$$

Since the above cases hold for any $s, a$, we therefore have:

$$\|Q_{k+1}^{UB} - Q_{k+2}^{UB}\|_\infty \leq \gamma \|Q_k^{UB} - Q_{k+1}^{UB}\|_\infty \tag{18}$$

Since the distance decreases by gamma with every iteration, it will converge to 0 and hence $Q^{UB}$ converges to a fixed point.

**2) Lower Bound**

The operator $\mathcal{T}_{\max}$ for the lower bound is defined as follows:

$$Q_{k+1}^{LB}(s, a) = (\mathcal{T}_{\max} Q_k^{LB})(s, a) = \max\left(Q_k^{LB}(s, a), \min_{s' \in \hat{T}(\cdot|s,a)}\left[\mathcal{R}(s, a, s') + \gamma \max_{a'} Q_k^{LB}(s', a')\right]\right) \tag{19}$$

$\hat{T}(\cdot|s, a)$ denotes reachable states from $s, a$.

We consider the change of difference between Q values between before and after the modified Bellman update (i.e., the difference between $\left|Q_k^{LB}(s, a) - Q_{k+1}^{LB}(s, a)\right|$ and $\left|Q_{k+1}^{LB}(s, a) - Q_{k+2}^{LB}(s, a)\right|$):

**Case 1:** If the first elements in max were the bigger values for both $Q_{k+1}^{LB}$ and $Q_{k+2}^{LB}$:

$$Q_{k+1}^{LB}(s, a) = Q_k^{LB}(s, a)$$

$$Q_{k+2}^{LB}(s, a) = Q_{k+1}^{LB}(s, a)$$

$$\left|Q_{k+1}^{LB}(s, a) - Q_{k+2}^{LB}(s, a)\right| = |Q_k^{LB}(s, a) - Q_{k+1}^{LB}(s, a)| = 0$$

**Case 2:** If the second element in max was the bigger value for $Q_{k+1}^{LB}$ and the first element in max was the bigger value for $Q_{k+2}^{LB}$:

$$Q_{k+1}^{LB}(s, a) = \min_{s' \in \hat{T}(\cdot|s,a)}\left[\mathcal{R}(s, a, s') + \gamma \max_{a'} Q_k^{LB}(s', a')\right]$$

$$Q_{k+2}^{LB}(s, a) = Q_{k+1}^{LB}(s, a)$$

$$\left|Q_{k+1}^{LB}(s, a) - Q_{k+2}^{LB}(s, a)\right| = 0$$

**Case 3:** If the first element in max was the bigger value for $Q_{k+1}^{LB}$ and the second element in max was the bigger value for $Q_{k+2}^{LB}$:

$$Q_{k+1}^{LB}(s, a) = Q_k^{LB}(s, a) \geq \min_{s' \in \hat{T}(\cdot|s,a)}\left[\mathcal{R}(s, a, s') + \gamma \max_{a'} Q_k^{LB}(s', a')\right] \tag{20}$$

$$Q_{k+2}^{LB}(s, a) = \min_{s' \in \hat{T}(\cdot|s,a)}\left[\mathcal{R}(s, a, s') + \gamma \max_{a'} Q_{k+1}^{LB}(s', a')\right]$$

$$\left| Q_{k+1}^{LB}(s,a) - Q_{k+2}^{LB}(s,a) \right|$$

$$= - \left( Q_k^{LB}(s,a) - \min_{s' \in \hat{T}(\cdot|s,a)} \left[ \mathcal{R}(s,a,s') + \gamma \max_{a'} Q_{k+1}^{LB}(s',a') \right] \right)$$

$$\text{(since } Q_{k+2}^{LB}(s,a) \geq Q_{k+1}^{LB}(s,a) \text{ based on Eq. 19)}$$

$$\leq - \left( \min_{s' \in \hat{T}(\cdot|s,a)} \left[ \mathcal{R}(s,a,s') + \gamma \max_{a'} Q_k^{LB}(s',a') \right] - \min_{s' \in \hat{T}(\cdot|s,a)} \left[ \mathcal{R}(s,a,s') + \gamma \max_{a'} Q_{k+1}^{LB}(s',a') \right] \right) \text{ (Eq. 20)}$$

$$\leq \left| \min_{s' \in \hat{T}(\cdot|s,a)} \left[ \mathcal{R}(s,a,s') + \gamma \max_{a'} Q_k^{LB}(s',a') \right] - \min_{s' \in \hat{T}(\cdot|s,a)} \left[ \mathcal{R}(s,a,s') + \gamma \max_{a'} Q_{k+1}^{LB}(s',a') \right] \right|$$

$$\leq \gamma \max_{s' \in \hat{T}(\cdot|s,a)} \left| \max_{a'} Q_k^{LB}(s',a') - \max_{a'} Q_{k+1}^{LB}(s',a') \right| \quad \text{(Lemma A.2)}$$

$$\leq \gamma \max_{s' \in \hat{T}(\cdot|s,a)} \max_{a'} \left| Q_k^{LB}(s',a') - Q_{k+1}^{LB}(s',a') \right| \quad \text{(Lemma A.1)}$$

$$\leq \gamma \| Q_k^{LB}(s,a) - Q_{k+1}^{LB}(s,a) \|_\infty$$

**Case 4:** If the second elements in max were the bigger values for both $Q_{k+1}$ and $Q_{k+2}$:

$$Q_{k+1}^{LB}(s,a) = \min_{s' \in \hat{T}(\cdot|s,a)} \left[ \mathcal{R}(s,a,s') + \gamma \max_{a'} Q_k^{LB}(s',a') \right]$$

$$Q_{k+2}^{LB}(s,a) = \min_{s' \in \hat{T}(\cdot|s,a)} \left[ \mathcal{R}(s,a,s') + \gamma \max_{a'} Q_{k+1}^{LB}(s',a') \right]$$

$$\left| Q_{k+1}^{LB}(s,a) - Q_{k+2}^{LB}(s,a) \right|$$

$$= \left| \min_{s' \in \hat{T}(\cdot|s,a)} \left[ \mathcal{R}(s,a,s') + \gamma \max_{a'} Q_k^{LB}(s',a') \right] - \max_{s' \in \hat{T}(\cdot|s,a)} \left[ \mathcal{R}(s,a,s') + \gamma \max_{a'} Q_{k+1}^{LB}(s',a') \right] \right|$$

$$\leq \gamma \| Q_k^{LB}(s,a) - Q_{k+1}^{LB}(s,a) \|_\infty \quad \text{(similar to Case 3)}$$

Since the above cases hold for any $s, a$, we therefore have:

$$\| Q_{k+1}^{LB} - Q_{k+2}^{LB} \|_\infty \leq \gamma \| Q_k^{LB} - Q_{k+1}^{LB} \|_\infty \tag{21}$$

Since the distance decreases by gamma with every iteration, it will converge to 0 and hence $Q^{LB}$ converges to a fixed point. □

**Theorem 3.8** The Bellman operator in M-Q-M specifies only a non-strict contraction in general:

$$\left\|\mathcal{T}Q - \mathcal{T}\widehat{Q}\right\|_\infty \leq \left\|Q - \widehat{Q}\right\|_\infty$$

*Proof.* 1) For $\mathcal{T}_{\min}$ computing the upper bound:

$$\left|\mathcal{T}_{\min}Q(s,a) - \mathcal{T}_{\min}\widehat{Q}(s,a)\right| = \left| \min\left( Q(s,a), \max_{s' \in \hat{T}(\cdot|s,a)} \left[ \mathcal{R}(s,a,s') + \gamma \max_{a'}(Q(s',a')) \right] \right) \right.$$

$$\left. - \min\left( \widehat{Q}(s,a), \max_{s' \in \hat{T}(\cdot|s,a)} \left[ \mathcal{R}(s,a,s') + \gamma \max_{a'}(\widehat{Q}(s',a')) \right] \right) \right|$$

$$\leq$$

$$\max\left( \left| Q(s,a) - \widehat{Q}(s,a) \right|, \right.$$

$$\left| \max_{s' \in \hat{T}(\cdot|s,a)} \left[ \mathcal{R}(s,a,s') + \gamma \max_{a'}(Q(s',a')) \right] \right.$$

$$\left. \left. - \max_{s' \in \hat{T}(\cdot|s,a)} \left[ \mathcal{R}(s,a,s') + \gamma \max_{a'}(\widehat{Q}(s',a')) \right] \right| \right) \qquad \text{(Lemma A.2)}$$

$$\leq$$

$$\max\left( \left| Q(s,a) - \widehat{Q}(s,a) \right|, \right.$$

$$\left. \gamma \left| \max_{s' \in \hat{T}(\cdot|s,a)} \max_{a'} \left[ Q(s',a') - \widehat{Q}(s',a') \right] \right| \right) \qquad \text{(Lemma A.1)}$$

$$\leq \max\left( \left\|Q - \widehat{Q}\right\|_\infty, \gamma \left\|Q - \widehat{Q}\right\|_\infty \right)$$

$$= \left\|Q - \widehat{Q}\right\|_\infty$$

2) For $\mathcal{T}_{\max}$ computing the lower bound:

$$
\begin{aligned}
\left| \mathcal{T}_{\max} Q(s,a) - \mathcal{T}_{\max} \widehat{Q}(s,a) \right| &= \left| \max \left( Q(s,a), \min_{s' \in \widehat{T}(\cdot|s,a)} \left[ \mathcal{R}(s,a,s') + \gamma \max_{a'}(Q(s',a')) \right] \right) \right. \\
&\quad \left. - \max \left( \widehat{Q}(s,a), \min_{s' \in \widehat{T}(\cdot|s,a)} \left[ \mathcal{R}(s,a,s') + \gamma \max_{a'}(\widehat{Q}(s',a')) \right] \right) \right| \\
&\leq \\
&\max \left( \left| Q(s,a) - \widehat{Q}(s,a) \right|, \right. \\
&\quad \left| \min_{s' \in \widehat{T}(\cdot|s,a)} \left[ \mathcal{R}(s,a,s') + \gamma \max_{a'}(Q(s',a')) \right] \right. \\
&\quad \left. \left. - \min_{s' \in \widehat{T}(\cdot|s,a)} \left[ \mathcal{R}(s,a,s') + \gamma \max_{a'}(\widehat{Q}(s',a')) \right] \right| \right) \quad \text{(Lemma A.1)} \\
&\leq \\
&\max \left( \left| Q(s,a) - \widehat{Q}(s,a) \right|, \right. \\
&\quad \left. \gamma \left| \max_{s' \in \widehat{T}(\cdot|s,a)} \max_{a'} \left[ Q(s',a') - \widehat{Q}(s',a') \right] \right| \right) \quad \text{(Lemma A.2)} \\
&\leq \max \left( \left\| Q - \widehat{Q} \right\|_{\infty}, \gamma \left\| Q - \widehat{Q} \right\|_{\infty} \right) \\
&= \left\| Q - \widehat{Q} \right\|_{\infty}
\end{aligned}
$$

Since the above holds for any $s, a$ and for both $\mathcal{T}_{\min}$ and $\mathcal{T}_{\max}$, we have the conclusion holds. □

**Theorem 3.11** [Optimality] For reward adaptation with Q variants, the optimal policies in the target domain remain invariant under Q-M or M-Q-M (under their ideal settings discussed, respectively) when the upper and lower bounds are initialized correctly.

*Proof.* Let

$$
\begin{aligned}
A_p(s) &= \{\widehat{a}| \; \exists a \; Q^{LB}(s,a) > Q^{UB}(s,\widehat{a}); a \neq \widehat{a}\} \\
\tilde{A}(s) &= A(s) \setminus A_p(s)
\end{aligned}
$$

where $A_p(s)$ represents the set of pruned actions under set $s$ and $\tilde{A}$ represents the remaining set of actions. To retain all optimal policies, it must be satisfied that none of the optimal actions under each state are pruned.

Assuming that a pruned action $\widehat{a}$ under $s$ is an optimal action, we must have

$$
\forall a \; Q^*(s,a) \leq Q^*(s,\widehat{a})
$$

Given that Q-M only prunes an action $\widehat{a}$ under $s$ when $\exists a \; Q^{LB}(s,a) > Q^{UB}(s,\widehat{a})$, we can derive that

$$
Q^{LB}(s,a) > Q^{UB}(s,\widehat{a}) \geq Q^*(s,\widehat{a}) \geq Q^*(s,a),
$$

resulting in a contradiction that

$$
Q^{LB}(s,a) > Q^*(s,a)
$$

As a result, we know that all optimal actions and hence policies are retained. □

**Corollary** 3.9 [Non-uniqueness] The fixed point of the iteration process in M-Q-M may not be unique.

*Proof.* This can be proved using the following example:

Consider a three state MDP with states s1, s2, s3, where from s1 agent can take an action that transitions uniformly (0.5) to s2 and s3, from s2 agent can take an action that transitions uniformly (0.5) to s1 and s3, and s3 is the terminal state. Reward is 1 for both actions. There is no reward for the terminal state. Assuming a discount factor of 0.5.

For the upper bound, depending on how V(s3) is initialized, it may result in different fixed points:

- When V(s3) is initialized to a big value (say 4), a fixed point may be V(s1) = 3 and V(s2) = 3;

- When V(s3) is initialized to a small positive value (say 1), another fixed point could be V(s1) = 3/2 and V(s2) =3/2.

$\square$

## A.2  Algorithm

---
**Algorithm 1** Reward Adaptation via Q-Manipulation
---
1: Retrieve variants of $Q$, reachable states, and source reward functions from source domains.
2: Initialize $Q^{UB}$ and $Q^{LB}$ for the target behavior.
3: Tighten the bounds using the iteration process in Q-M or M-Q-M.
4: Prune actions.
5: Perform learning in the target domain with the remaining actions.
---

Github URL: https://github.com/kevin-jatin-vora/Action-pruning-in-Reward-Adaptation

## A.3  Domain Information

### A.3.1  MDP Generation

For autogenerated MDP creation: the transitions and transition distributions are then randomly generated. Initially, the number of reachable states from any $s, a$ is $|A| = 9$. However, when an SBF is set for the generated MDP: for each $s, a$ pair, 1) we first randomly select a number $k$ from [1, SBF] as the number of reachable states from $s, a$, 2) we retain the state from the transition with the highest probability (which is often the "intended" state) while randomly choosing $k - 1$ states (without replacement) from its remaining reachable states; these are then considered as the new reachable states from $s, a$, and 3) re-normalize the transition distribution for $s, a$ based on these new reachable states. 3 states are randomly chosen to be the terminal states. Note that a new MDP is generated for each run. Similarly, for gridworld domains, in each run, the MDP is slightly different with respect to random SBF updated in a manner analogous to the auto-generated MDPs. For our implementation, we resort to the most general form of reward function $R(s, a, s')$.

Detailed descriptions of the domains used for our evaluations are given below:

**Dollar-Euro:** A 45 states and 4 actions grid-world domain as illustrated in Fig. 1. **Source Domain 1 with $R_1$ (collecting dollars):** The agent obtains a reward of 1.0 for reaching the location labeled with "\$", and 0.6 for reaching the location labeled with both \$ and €. **Source Domain 2 with $R_2$ (collecting euros):** The agent obtains a reward of 1.0 for reaching the location labeled with €, and 0.6 for reaching the location labeled with both \$ and €. **Target Domain with $\mathcal{R}$:** $\mathcal{R} = R_1 + R_2$.

**Frozen Lake:** A standard toy-text environment with 36 states and 4 actions. An episode terminates when the agent falls into any hole in the frozen lake (4 holes in total) or reaches the goal. **Source Domain 1 with**

$R_1$**:** The agent is rewarded $+1$ for reaching any hole in a subset of holes (denoted by $H$), $-1$ for reaching any hole in the remaining holes (denoted by $\widehat{H}$) and 0.5 for reaching the goal. **Source Domain 2 with** $R_2$**:** The agent is rewarded $+1$ for reaching any hole in $\widehat{H}$, $-1$ for reaching any hole in $H$, and 0.5 for reaching the goal. **Target Domain with** $\mathcal{R}$**:** Avoid all the holes and reach the goal, or $\mathcal{R} = R_1 + R_2$.

**Race Track:** A 49 states and 7 actions grid-world domain. The 7 actions correspond to different velocities for going forward, turning left, or turning right. An initial location, a goal location, and obstacles make up the race track. An episode ends when the agent reaches the goal position, crashes, or exhausts the total number of steps. **Source Domain 1 with** $R_1$ **(avoid obstacles):** The agent obtains a negative reward of $-0.5$ for collision with a living reward of $+0.2$. **Source Domain 2 with** $R_2$ **(terminate):** The agent obtains a reward of $+2$ for reaching the goal, $-0.3$ living reward, and $-4$ for staying at the initial location. **Source Domain 3 with** $R_3$ **(stay put):** The agent obtains a reward of $+3$ for staying at the initial location. **Target Domain with** $\mathcal{R}$**:** Reach the goal in the least number of steps while avoiding all obstacles, or $\mathcal{R} = R_1 + R_2 + R_3$. This is the only domain where there are three source behaviors.

**Auto-generated Domains:**

Generate MDPs with the number of actions=9 and the number of states =60. Rewards for the source domains (i.e., $(R_1, R_2)$) for two of those states are set to $(+1, -1)$ and $(-1, +1)$, respectively; rewards for the third terminal state are set to $(+0.6, +0.6)$.

## A.4 Learning with Q-M in Practice

It is important to note that Q-variants may be difficult to learn with the same samples as experienced during a typical Q-learning process for $Q^*$. Some adaptation to Q learning must be made in order to learn $Q^*$ and $Q^\mu$ (or other Q-variant) via the same set of samples. Note that theoretically, Q learning is guaranteed to converge regardless of the behavior policy, although that is inefficient and can result in inaccuracy in practice due to that the behavior policy may result in visiting a different distribution of the states from that of the optimal policy (distributional shift). To ensure that $Q^*$ and $Q^\mu$ (or other Q-variant) can both receive informative samples, one possible way is to alternate between training $Q^*$ and $Q^\mu$ (or other Q-variant) and use importance sampling while using samples from $Q^\mu$ (or other Q-variant) to training $Q^*$ (or vice versa), so that we can leverage samples from both $Q^*$ and $Q^\mu$ (or other Q-variant) to train both $Q^*$ and $Q^\mu$ (or other Q-variant). Fig. 6 shows that approximately the same number of samples are used in the training of individual behaviors for the autogenerated MDP. Moreover, for learning in practice, we rely on memorizing the reward functions during the learning of source behaviors. Since the bounds are approximate and pruning may result in no actions remaining for a given state in practice, we rely on a fallback mechanism that ensures the highest-value action within the upper bound is never pruned. Henceforth, we use Q-M$_p$ to represent Q-M and M-Q-M$_p$ to represent M-Q-M with memorized reward and reachable state functions while learning the source. Since Q-M does not depend on source Q values, memorized reward can be same as true reward and reachable state may also be memorized accurately, Q-M$_p$ performs similar to Q-M. In such cases we represent it as Q-M (Q-M$_p$) to avoid cluttering. As a result, Q-M computes bounds using both the learned value functions and the memorized rewards, which has been shown to work well in practice (Fig. 6 and 7). For the racetrack domain, we observe that optimal actions are pruned, leading to suboptimal convergence as SBF increases. We believe this is due to the approximate estimate of Q in the source domain, which results in incorrect initialization and subsequently pruning of optimal actions due to inaccurate bounds. In such cases, Q-M serves as a better alternative since it does not rely on Q-value initialization.

### A.4.1 Hyperparameters

All hyperparamters are set to be same for the different methods in the same evaluation domain. The exploration rate starts from 1.0 and is gradually decayed. $\gamma$ is chosen between $[0.9, 0.99]$ across different domains.

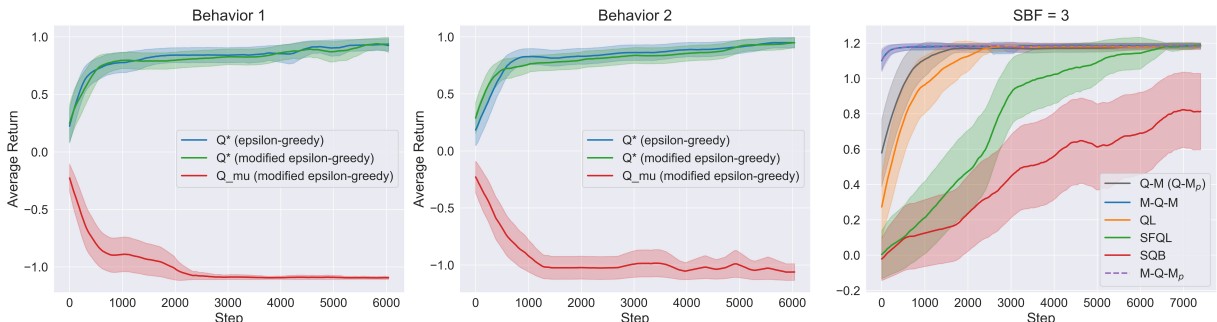

Figure 6: Convergence plot for autogenerated domain with linear reward combination: Behavior 1 (left), Behavior 2 (center) and Target (right) where M-Q-M$_p$ performs action pruning using a estimated lite model, reward model, and Q-variants. M-Q-M$_p$ indicates M-Q-M in practice.

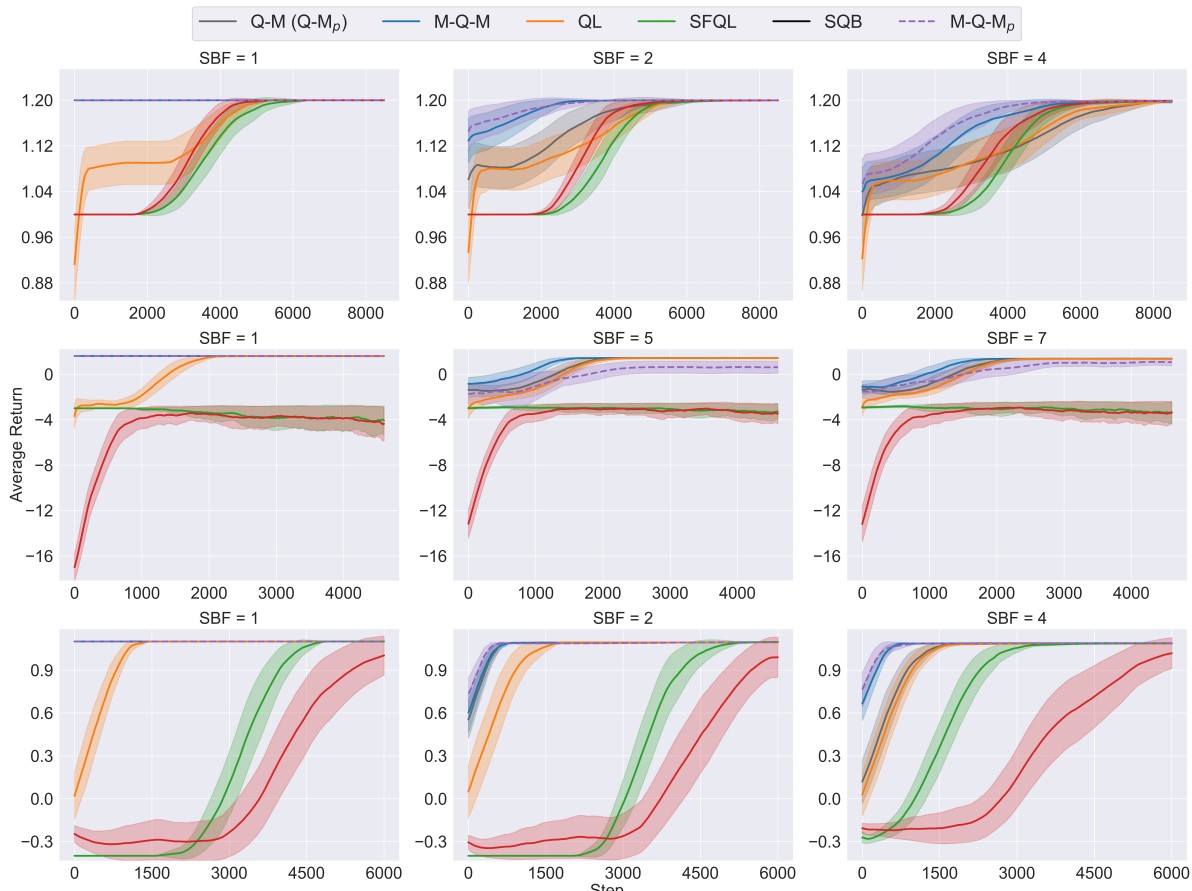

Figure 7: Convergence plot for: (top) Dollar Euro, (middle) Racetrack and (bottom) Frozen Lake using a estimated lite model, reward model, and Q-variants. M-Q-M$_p$ indicates M-Q-M in practice.

### A.4.2 Running Time Comparison

We measured the running times taken for overhead cost of computing bounds and pruning actions for each method on an XPS 9500 laptop. The aim here is to show that Q-M iteration adds, in most cases, a reasonable amount of extra computation to the entire learning process (refer Table 2).

| Domain | SBF_min | SBF_mid | SBF_max |
|---|---|---|---|
| Dollar Euro | 0.05 | 0.04 | 0.03 |
| Race Track | 0.13 | 0.12 | 0.15 |
| Frozen Lake | 0.04 | 0.04 | 0.04 |
| Autogenerated | 0.11 | 0.11 | 0.12 |
| Non-linear Target Reward | 0.23 | 1.2 | 0.42 |
| Noisy Reward Combination | 0.12 | 0.12 | 0.13 |

Table 2: Running time for Q-M iteration process (in seconds) by Domain and SBF

# B   Additional Results

## B.1   Action pruning Analysis

While heatmaps in Figure 3, only show the number of actions pruned, Figure 8 gives a deeper insight into which actions are pruned. Note that as a consequence of this pruning, some states may become unreachable from a given state. This is by design since those states do not contribute to the optimal behavior if the optimal behavior passes through the given state. The values for those states may not be updated properly during target training but as long as the initial state is chosen consistently during training and testing, it should not pose any problem.

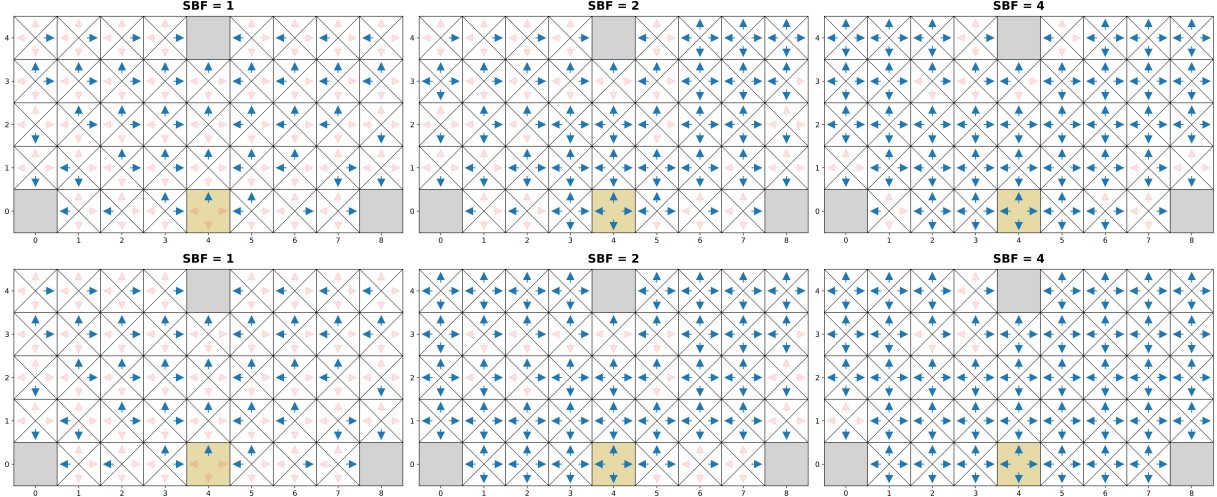

Figure 8: Dollar Euro action pruning: Red = Pruned, Blue = Retained. Top: M-Q-M, Bottom: Q-M. Note that action pruning is not symmetric due to our random construction of the transition function (see Sec. 4.2)

## B.2   Pruning Threshold

In section 3.3, we provide a way to set the Delta parameter which ensures pruning only happens when the Lower bound of some action $a$ is at least $\Delta$ units greater than the upper bound of another action $a_{hat}$. While this ensures optimal action is never pruned, $\Delta$ can also be set empirically or less conservatively. The conservative value of $\Delta$ as per section 3.3 is $1e-11$ and optimal actions are not pruned till $\Delta = 0$. Setting a lower $\Delta$ may lead to pruning more actions, and sometimes it is at the cost of a loss of optimal action (refer Figure 9). In our evaluation to study the varying effect of delta, we find that delta parameter is robust, and optimal actions are pruned if delta is set to a negative value. As shown Figure 9, the red count in the bar indicates the number of actions which were optimal but still pruned due to the non-conservative value of $\Delta$. Setting delta is domain dependent to some extent, and so it should be set conservatively to ensure optimality is not lost.

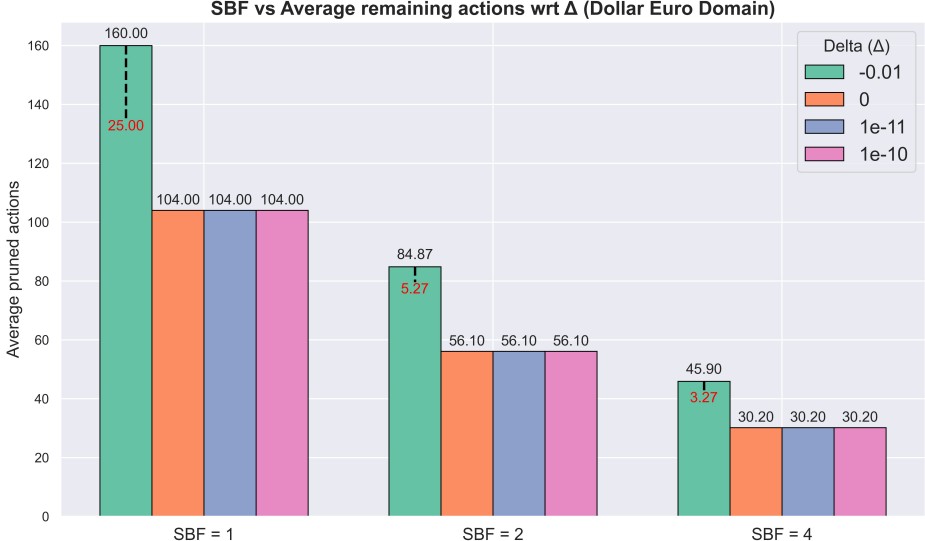

Figure 9: Impact of $\Delta$ on the number of actions pruned in the Dollar Euro domain; count in red indicates instances where an optimal actions were pruned.

### B.3 Noisy Source $R_i$

In Section A.4, when learning the reward function and Q in practical scenarios, memorization is beneficial when the reward is deterministic. However, when the reward in the source domain is noisy, we employ an exponential moving average to approximate the reward function. For our experiments, we consider the addition of uniform noise to the source behavior during source behavior training (given by Equation (22)). Since $R$ may deviate from the true source reward, the Q-values learned from the source domain could yield inaccurate bounds, potentially leading to the loss of the optimal solution. We implemented a fallback mechanism to not prune actions in states where LB>UB (due to noisy reward or approximate value function), but this does not ensure optimality under a noisy reward function (as shown in Figure 10). For M-Q-M, if initialization is incorrect, then it may never recover from incorrect bounds, which may lead to pruning of optimal actions. For Q-M, if the mean reward is accurate (which can be ensured through sufficient exploration during source domain training), optimal actions should not be pruned. We know that as SBF increases the number of actions pruned decreases. Furthermore, stochastic noise in reward for one state may offset noisy reward in other states. As a result we might not see pruning of optimal actions but very few actions are pruned which results into QL like performance. In the future, we plan to study this effect further and establish safeguards against pruning out of optimal actions.

$$\hat{R}_i = R_i + U(+x, -x) \tag{22}$$

### B.4 Scalability

Q-M and M-Q-M are expected to scale analogously to value iteration due to similarity in the updates. In order to verify this, we perform evaluations on scaled version of the Dollar-Euro domain when SBF=2. As shown in Figure 11, Q-M and M-Q-M outperform other baselines. It is important to note that, scaling in environment size affects baseline more than Q-M and M-Q-M. It is observed that our method significantly reduces training time with respect to the baselines which becomes more evident as state space grows larger. Table 3,4,5 presents a detailed breakdown of the time required for source training using QL (see Sec. A.4), the overhead associated with computing bounds and pruning actions, and the total training time. All times are reported until the agent achieves 95% of the maximum reward during both source and target training. Q-M does not require source Q-values and so source training time only includes time taken for memorizing reward and reachability function. Figure 12 illustrates action pruning under M-Q-M, where the source Q-values are

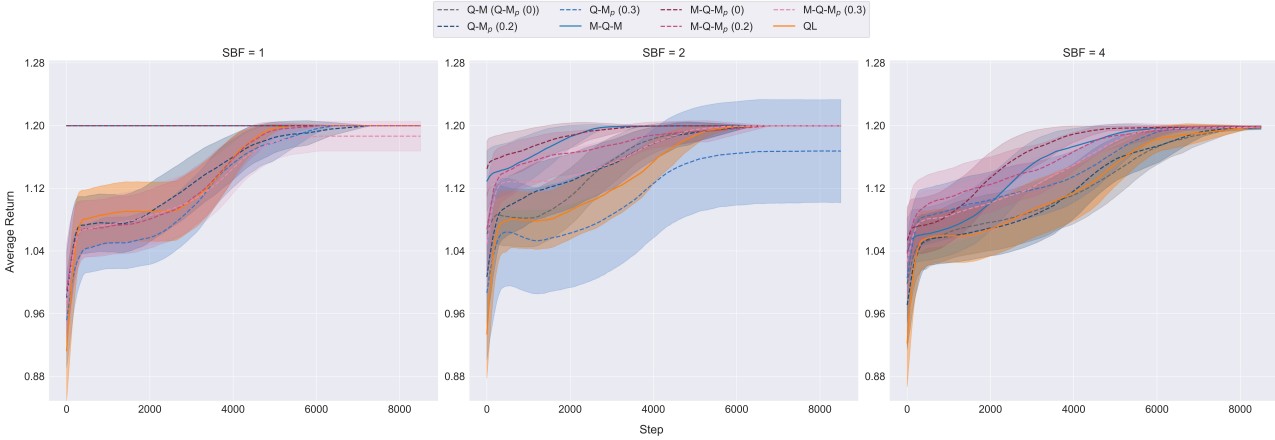

Figure 10: Convergence plot for Dollar Euro domain when source behaviors have noisy reward indicated within parentheses of M-Q-M$_p$

learned using Q-Learning and value iteration (which is employed in the main paper for evaluating the theory). It is important to note that our primary aim is to improve sample efficiency and source behaviors, source rewards, the combination function, and the lite model are assumed available to all approaches.

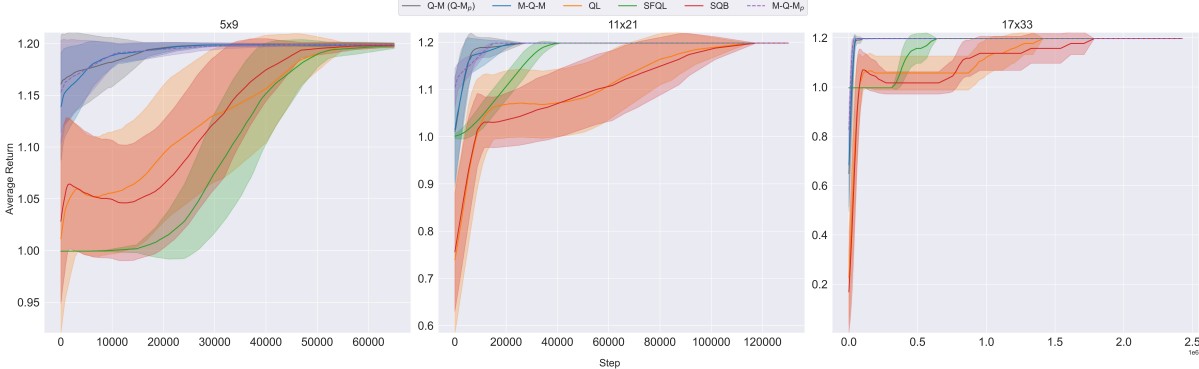

Figure 11: Convergence plot for Dollar Euro Domain 5x11, 11x21, 17x33 (left to right)

| Method | Source Training | | Overhead | Target training | | Total | |
|--------|------|-----|----------|------|-----|------|-----|
| | mean | std | | mean | std | mean | std |
| M-Q-M$_p$ | 0.25 | 0.34 | 0.04 | 0.01 | 0.01 | 0.34 | 0.35 |
| Q-M$_p$ | $1e^{-4}$ | $1e^{-5}$ | 0.27 | 0.05 | 0.05 | 0.32 | 0.05 |
| QL | - | - | - | 0.08 | 0.07 | 0.08 | 0.07 |
| SFQL | 0.2 | 0.14 | * | 0.08 | 0.05 | 0.28 | 0.19 |
| SQB | | | - | 0.05 | 0.09 | 0.25 | 0.23 |

Table 3: Running time (in seconds) for Dollar Euro 5x9

| Method | Source Training | | Overhead | Target training | | Total | |
|--------|------|------|----------|------|------|------|------|
| | mean | std | | mean | std | mean | std |
| M-Q-M$_p$ | 1.82 | 0.89 | 0.04 | 0.79 | 0.41 | 2.65 | 1.3 |
| Q-M$_p$ | 0.07 | 1e$^{-5}$ | 0.29 | 0.8 | 0.68 | 1.16 | 0.68 |
| QL | - | - | - | 2.25 | 1.96 | 2.25 | 1.96 |
| SFQL | 1.51 | 0.95 | * | 1.27 | 0.23 | 2.78 | 1.18 |
| SQB | | | - | 3.26 | 2.94 | 4.77 | 3.89 |

Table 4: Running time (in seconds) for Dollar Euro 11x21

| Method | Source Training | | Overhead | Target training | | Total | |
|--------|------|------|----------|------|------|------|------|
| | mean | std | | mean | std | mean | std |
| M-Q-M$_p$ | 8.67 | 3.43 | 0.05 | 2.23 | 1.75 | 10.95 | 5.18 |
| Q-M$_p$ | 0.69 | 1e$^{-4}$ | 0.31 | 4.31 | 2.31 | 5.31 | 2.31 |
| QL | - | | - | 28.46 | 44.02 | 28.46 | 44.02 |
| SFQL | 8.69 | 4.81 | * | 37.02 | 50.55 | 45.71 | 55.36 |
| SQB | | | - | 62.28 | 87.47 | 70.97 | 92.28 |

Table 5: Running time (in seconds) for Dollar Euro 17x33

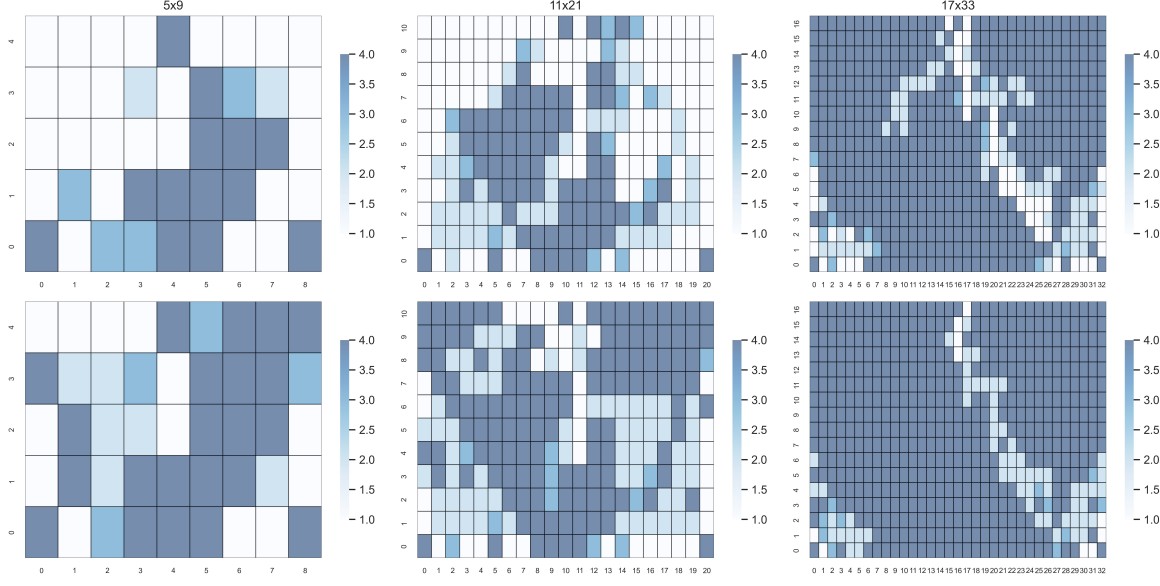

Figure 12: Heatmap for action remaining in each state after pruning using M-Q-M for Dollar Euro Domain (top: source behaviors trained using value iteration; bottom: source behaviors trained using the modified QL (refer to A.4))

