# OpenReview forum: "Provably Efficient Reward Transfer in Reinforcement Learning with Discrete Markov Decision Processes"
_TMLR — Accepted by TMLR_

### Review · Reviewer_yo5a · 2025-11-17

**Summary Of Contributions:**

### Summary:

This paper proposes a novel approach to reward adaptation in reinforcement learning. The core idea is to use a lite-model of the system dynamics learned from source domains to compute upper and lower bounds for the Q-values of different actions. The important assumption is that the dynamics of the system, and the lite-model as a result, are shared between source and target domains. These bounds are then used to prune actions at different states in the target task. This action pruning increases the efficiency of Q-learning in the target domain. To compute the bounds, the authors propose two variants: Q-Manipulation (Q-M) and Monotonic Q-Manipulation (M-Q-M). The main difference between these methods is that Q-M may lead to loose bounds, whereas M-Q-M produces tighter bounds. The issue with M-Q-M is that this method requires a somewhat accurate initialization for the bounds. This initialization can be calculated exactly in some specific and limited cases listed in the paper. M-Q-M guarantees that the upper bounds are monotonically non-increasing and the lower bounds are monotonically non-decreasing. The authors compare Q-M variants against Q-Learning, Successor Feature Q-Learning (SFQL) [1], and Soft Q Bounding (SQB) [2]. M-Q-M and Q-M outperform baselines across various domains and levels of stochasticity. However, when the function relating source and target rewards is noisy, the methods, especially Q-M, do not provide meaningful gains compared to Q-Learning.


### Strengths:
1. The methods outperform baselines across various tasks and levels of stochasticity in terms of sample efficiency during learning in the target domain, and they outperform SFQL and SQB on racetrack tasks in final performance.
2. They do not require similarity between source and target domains. In fact, rewards may be contradictory.
3. They are robust to negative transfer.
4. The paper provides theoretical guarantees.
5. The problem is well defined.
6. The problem is clearly defined, and the experimental setup is well documented.
7. Results are reproducible via the anonymized code repository.
8. Limitations of the method are clearly mentioned.


### Weaknesses:

   1. As the authors acknowledge, applicability of this method in the current form to more complex scenarios is limited.

   2. In many cases, access to a lite-model is not feasible, even in discrete state spaces, and the method relies heavily on that model.

   3. The paper does not explain how the lite model can be practically obtained during learning source domains. Also, it is not studied how sensitive the method is to errors in that model. No ablation is provided for incomplete or partially learned lite models (e.g., when some reachable states are unseen). In large state spaces, especially with online model estimation or many terminal states, the learned model may reflect policy-dependent blind spots.
   4. M-Q-M relies on proper initialization, and the proposed method for computing this initialization is only applicable when the relationship between the source and target domains is exactly known and rewards are deterministic. There is no discussion of cases where rewards are stochastic. My expectation is that these methods would fail due to maximization/minimization bias.
5. No ablation study is provided on the effect of varying delta (the allowed gap between the upper bound of one action and the lower bound of others to not be pruned). This factor is extremely important to set correctly, so such an ablation is essential.
6. Safe action pruning requires having access to some factors which as mentioned by the authors is not available; therefore, we can never find the optimum delta and the method may prune some safe and important actions.
7. The time comparison with baselines is based only on sample efficiency when learning the target domain. Runtime is reported only for Q-M. It is unclear whether, after accounting for the time needed to obtain the lite model, compute proper initialization, and perform action pruning, Q-M remains justifiable, especially compared to SQB which performs online training.
8. The related work section is weak. Prior work cited by the SQB baseline (which is the most closely related method) includes many papers that are not discussed by the authors (see Section 3 of [2]).
9. In analyzing action pruning, the authors only show the degree of pruning across states. They do not specify which actions are pruned at each state, which is important for assessing whether the pruning is sensible.
10. Action pruning leads to safety concerns in scenarios in which we have some risky regions. More analysis is required to see if action pruning is safe. Especially in environments which have risky states (for example minigrid with lava [3]).
11. In some theories, the assumptions are not clearly mentioned. For example, the contraction proof used in Theorem 3.4 is correct only for deterministic MDP. However, this assumption is not mentioned in the body of the theorem. Or in Theorem 3.5 no assumption on the types of functions allowed to connect source and target domains is mentioned.


### References


[1] Barreto, André, et al. "Successor features for transfer in reinforcement learning." Advances in Neural Information Processing Systems 30 (2017).


[2] Adamczyk, J., Makarenko, V., Tiomkin, S., \& Kulkarni, R. V. (2024). Boosting Soft Q-Learning by Bounding. arXiv preprint arXiv:2406.18033.


[3] https://minigrid.farama.org/environments/minigrid/LavaGapEnv/

**Audience:**

Yes

**Audience Explanation:**

The results are promising and outperform some baselines in certain toy environments. The claims are supported by theoretical findings, which may interest those focused on the theoretical aspects of reinforcement learning. However, researchers interested in practical applications may find this method less useful, as it relies on several components and assumptions that hold only in limited or simplified environments.

**Broader Impact Concerns:**

In the section authors talking about the risk, they focus on the importance of choosing rewards for shaping initializations safely. However, the possible adverse effects of action pruning should be noted as well.

**Claims And Evidence:**

Yes

**Claims Explanation:**

All of the claims are supported by theory and experiments.

**Requested Changes:**

### For acceptance:

1. Authors need to add wall-clock time comparison with baselines. This time must include the time required for source domain training as well to be fair.
2. More analysis of action pruning. Showing pruned actions to see if pruning is meaningful or somewhat random, and pruning in more environments. I am especially interested to see how pruning is performed close to critical states (high/low reward)
3. All assumptions behind theorems should be mentioned clearly in the body of the theorems.
4. Experiments showing how the method scales by increasing the size of environments (just some managable increase in the size).
5. What will happen if the source domain rewards are noisy (not the function which maps source rewards to the target rewards)? Any difference between noisy source rewards and noisy function connecting source to target?
6. Ablation study on the effect of varying delta in action pruning should be added.

### Good to be added:

1. Any practical recipe for calculating lite-model?
2. If we need to estimate lite-model in an online way during source model training, will there be any blind spots? How much more compute do we need to remove those blind spots?
3. In the repository you have mentioned, I see some results regarding continuous case. Any success there would be interesting to see.

---

> ### Comment · Reviewer_yo5a · 2025-12-07
> **Review for the revised version posted on Nov 24, 2025**
>
> I would like to thank the authors for providing additional experiments. These new results address the main questions I previously raised. However, they also reveal some new ambiguities and potential weaknesses that I would like to discuss.
>
> Appendices:
>
> **B.1:**
> The results showing which actions are pruned reveal that, even in the simplest environment, the action pruning method can create blind spots. For example, in both methods with SBF=1, state (0,8)[which is an important terminal state] cannot be reached from state (2,7). Similar blind spots exist in other states as well.
> Question: Are these blind spots acceptable? If they could lead to issues, do the authors have suggestions on how to prevent them during action pruning?
>
> **B.2:**
> This appendix demonstrates that the proposed method scales well in terms of average return. However, it would be helpful to include a table summarizing the total wall-clock time (including source training, pruning, target training, etc.) across multiple environment sizes. This would provide a clearer picture of the computational complexity associated with scaling. In terms of performance, there does not appear to be any drop.
>
> **B.3 and B.4:**
> No further questions. The results are clear.
>
> **B.5:**
> The results remain somewhat ambiguous. It would be helpful to merge this appendix with B.2 and report the total wall-clock time (from the beginning of source domain training until reaching 95% of the maximum average return in the target domain) for all methods across several environment sizes (at least three). This would allow for a more comprehensive understanding of computational efficiency.
>
> Other parts:
> 1. Following the revision, the presentation of the theorems are now clear.
> 2. Authors added results regarding the case in which the source rewards are noisy. It seems that the proposed method **does not** outperform Q-Learning in this scenario.

---

> > ### Comment · Reviewer_yo5a · 2025-12-11
> > **Final Review (Dec. 10, 2025)**
> >
> > I would like to thank the authors for addressing the questions I raised in my previous response. This will be my final review, as I believe the paper has reached its limit in terms of possible improvements. My comments are based on the version submitted on Dec 10, 2025.
> >
> > **B1:** Unfortunately, it seems that one important and unresolvable limitation of this method, as confirmed by the authors in appendix B.1, is that it may create multiple blind spots or even infinite cycles, even in simple toy environments. The authors argue that if the initial state is fixed, this would not be a problem. However, I still believe that even in that case, the method can produce blind spots or infinite loops at any mid location, and there is no guarantee or specific way to handle these situations. This issue may become even more serious in more complex environments with critical states which we do or do not want to visit.
> >
> > **Total wall-clock time:** As I suggested, the authors have merged the appendices related to scaling and wall-clock time. The new appendix clearly shows how the proposed method performs well when the environment is scaled up. In the largest environment, the total wall-clock time of their best performing method (M-Q-M) is much smaller than all the other baselines (2.8x faster than QL, 4.5x faster than SFQL, and 7x times faster than SQB).

---

> > > ### Comment · Reviewer_KQD6 · 2026-02-22
> > > **Question about pruning safety**
> > >
> > > If reviewer y05a is still available for discussion, could you elaborate on the potential issues you see with pruning?
> > >
> > > If the assumptions are satisfied that T^ contains all p>0 transitions and we have accurate Q^* and Q^mu values, it seems expected to me the some portions of the space are no longer reachable in the pruned space for the final learning step, as some provably sub-optimal actions are removed.
> > >
> > > Is the reviewer's concern still present with this most-ideal version? This version seems correct to me, so I'd be curious to understand if I have missed something.
> > > Or does the concern come from the handling of a bound on approximated Q^* and Q^mu values? Or with learning or providing T^?

---

### Review · Reviewer_KYkx · 2025-11-30

**Summary Of Contributions:**

The work proposes Q-Manipulation (Q-M), a transfer learning framework for reward adaptation in discrete MDPs that significantly improves sample efficiency by safely pruning the action space. The proposed method iteratively computes upper and lower bounds on the target Q-function to eliminate suboptimal actions and does not sacrifices optimality. Ttheoretical guarantees and emperical experiments are included to further verify the effectiveness of the proposed methods.

**Audience:**

Yes

**Audience Explanation:**

RL is obviously an interesting topic for TMLR's audience. The contribution of Q-M can benefit the community.

**Broader Impact Concerns:**

The paper focuses on algorithmic improvements for sample efficiency in RL. The authors already briefly discuss relevant safety considerations regarding action pruning. I think this should be sufficient for this type of theoretical work.

**Claims And Evidence:**

Yes

**Claims Explanation:**

In general, I believe the claims in the submission are supported by accurate, convincing and clear evidence. The authors support their theoretical claims with formal proofs, demonstrating that the method converges. In addition, the pruning strategy is safe and will not discard optimal policies. Moreover, the empirical experiments further show the effectiveness of Q-manipulation.

**Requested Changes:**

1. I would like to see more discussion on computational overhead for a fair assessment of the trade-off between sample efficiency and computational cost. How the wall-clock time of the pre-computation phase scales with state space size?

2. Given that M-Q-M fixed points are not unique (Corollary 3.9),  please briefly discuss the practical implications.

3. Does the initialization strategy proposed reliably converge to "tight" bounds, or is there a risk of converging to loose bounds that offer no pruning benefit?

---

### Review · Reviewer_KQD6 · 2026-02-18

**Summary Of Contributions:**

This paper introduces Q-Manipulation (Q-M), an optimality-preserving transfer learning method using action pruning. The authors provide a proof of correctness, showing the iterative step generated correct upper and lower bounds, with correct pruning following immediately from that. Experimental results are provided for some synthetic domains, showing that pruning is happening in the subsequent learning phase, and does speed up learning.

As a strength, Q-M is uncomplicated for an optimality preserving method.

While the early sections of the article were clear, I found the method and experimental section harder to follow, partly because of what I think are fairly minor notational issues.

The introduction and related work did describe the general problem setup, but I did not have a motivating problem in mind and the article did not provide one. Specifically, a problem that (1) is small enough that I can reasonably solve for Q^LB and Q^UB in the pre-processing step for a target domain, (2) is large enough that the speed-up from pruning matters, (3) has exact T^ available.
In short: is there a real problem where I could actually run Q-M, have guarantee of optimality, and expect to see a real improvement in total training time (counting pre-processing)?

**Audience:**

Yes

**Audience Explanation:**

Transfer learning is an active research area, and in problems where Q-M's requirements are satisfied, it provides a combination of preserving optimality and the ability to get some speedup out of source reward data that is dissimilar to the target rewards.

**Claims And Evidence:**

Yes

**Claims Explanation:**

I believe the main theoretical claims are correct.

While the domains are all (arguably?) small, and sometimes synthetic, the experimental results are mostly complete: the authors look at performance with a combination of a few domains and a few different types of source->target reward functions. There is some analysis giving some extra evidence that pruning behaves as expected across some different domain characteristics. The authors do give some consideration to the processing time.
The description of the experiments is clear: I believe I understand the setup well enough to replicate it.

**Requested Changes:**

Regarding T^
abstract: “assumes access to a lite-mode, which is easy to provide or learn”.
introduction: “lite-model of the transition function that is easier to provide or estimate”
...
definition 3.3: “An assumption here is that each transition must have been experienced at least once by at least once by at least one of the source domains”
The phrase “easy to learn” without any other caveats suggests to me that I could almost certainly do it. Having to experience every transition at least once in source data seems impractical for many (larger?) problems. However, this is actually needed to preserve optimality, one of the paper’s main claims.
Some more care is needed in the statements in the abstract and introduction. Q-M is optimal, with an exact T^. T^ can be estimated, but Q-M might not preserve optimality, without learning an exact T^.

---
Experimental  results:

“To demonstrate the influence of SBF, for each evaluation domain, we gradually increase its SBF.”
What do the authors mean here? Just that experiments are done on a range of SBF values? Or is there actually a change over time during training?

“In the appendix, we use memorization and learning to estimate these from source domains, which demonstrate comparable performances.”
Comparable to what? Restate to be explicit: “and these variants have comparable performance to their exact counterparts.”

“We first evaluate with gridworld domains where combination is a linear combination” where target R is a linear combination?

Are the steps on the x axis of Figure 2 a roughly equivalent amount of work across the different methods?

“running time taken by the Q-M iteration process are reported in Sec. A.3”
A.4.2?
What are the running times of the 6000 steps for each of the methods, once post-processing is finished?

---
Section 3.1.1 “via success features” -> successor

Definition 3.2: what are R_1, … R_n?  If these are from n source domains, possibly use a sentence like “The source domains M_1, M_2, …, M_n are no longer accessible” in the previous paragraph.

3.1.2 “typically leads to a smaller mixing time t_mix”
A guarantee of lower sample complexity requires a guarantee that the mixing time has not increased by a larger factor than the decrease in (SA)^2: should this statement be “leads to a mixing time t_mix that is no larger and typically smaller”?

Theorem 3.4: T’ -> T? Holds for both T_max and T_min?
Also, to more easily distinguish it from the restricted sequence contraction in 3.7, this should probably be the general ||TQ_Aa-TQ_B|| <= gamma ||Q_A-Q_B|| as used in the proof, rather than Q_k Q_k+1 (which suggests a sequence, not arbitrary Q)

3.2.2 “transferred from the source to the target domains, referred to as Q variants (Q^* and Q^mu).”  For clarity, it would be good to re-state the optimal / “worst” distinction here, and possibly introduce the _i subscript used later. As it is, this statement kind of looks like it’s saying that Q^* is to do with the source domain and Q^mu is to do with the target domain. Maybe something like “Computing such an initialization can be done by transferring the source domain Q^*_i and “worst” policy Q^mu_i values to the target domain.”

Theorem 3.8  M-Q-M, not Q-M

Nonlinear Combination Function: How does Q^*_|R_i| differ from Q^*_i as in the linear case right above? If there is actually a difference, it needs to be stated. If it’s just a different choice of notation, they should be using the same notation (and throughout the paper).

---

> ### Comment · Reviewer_KQD6 · 2026-02-18
>
> Followup. After hitting submit -- sorry -- I see that my comment about total run time is already addressed later in appendix B.

---

> > ### Author Response · Authors · 2026-02-20
> >
> > We express our gratitude to the reviewer for your valuable feedback and the time invested in assessing our work. Please see our responses below and changes in red in the updated manuscript:
> > * **Lite-Model $\hat{T}$:** We agree that it is important to more carefully state the claim regarding the litemodel. We have updated the abstract to state that “The iteration process is based on a lite-model, which is assumed to be given or can be learned” and “We formally prove that Q-M, under discrete domains and an accurate lite-model, does not affect the optimality of the returned policy”.  We have also updated the introduction to state that “This process operates on a lite-model of the transition function that is assumed to be provided and is accurate. To learn such a model accurately in practice when the model is not available, an assumption must be made that each transition is experienced at least once, which is a less stringent requirement than that of value convergence of RL”. Below Definition 3.3, we state that “Optimality is no longer guaranteed when such an assumption does not hold”. We also acknowledge the limitations of such a lite model under section 5 Limitations and Future Work (Scaling to Real-World Domains).
> >
> >
> >     **One more note:** In our work, we assume access to an accurate lite-model. To guarantee optimality, it only needs to be conservative, or more specifically, the lite-transition model must include all the valid transitions (invalid transitions may also be included).   Such a model, hence, can be specified more easily for systems with stochastic transitions that are range-bounded, i.e., the next state can be specified as a distribution with a known mean and maximum deviation. Of course, more accurate models generally provide more pruning opportunities.
> >
> > * **Experimental results:**
> >     * SBF: For each experiment, the SBF remains constant, and distinct experiments are conducted for various SBFs (SBF is not modified during training).
> >     * We have updated Section 4.2 Evaluation design to include that memorization leads to similar performance to its counterparts and “target R is a linear combination of the source rewards”.
> >     * Figure 2: The goal of Figure 2 is to showcase samples used to reach the convergence threshold after necessary pre-processing. Q-M and M-Q-M require pre-processing to compute bounds before learning in the target domain. Similarly, for SFQL, policy evaluation of the source behaviors is required to bootstrap target learning. These are all computational and hence do not affect the samples needed. Q-M and M-Q-M are more expensive computation-wise, but save samples. We also clarify this under the “pre-learning” category of Table 1. We also reported the preprocessing times in Section B.4. For both source and target training, we measure the time needed to reach 95% of the maximum observed average return instead of a fixed number of steps.
> >
> > * Updated “The source domains $M_1, M_2, …, M_n$ are no longer accessible” in the paragraph before Definition 3.2.
> > * Updated 3.1.2 to state “While it is not straightforward to theoretically establish changes to the mixing time of the MDP after pruning, as long as the simplified transition structure leads to a mixing time that is not larger, sample efficiency will improve”.
> > * Updated Theorem 3.4 to include Q_A,Q_B instead of Q_k and Q_{k+1}.
> > * Updated 3.2.2 to restate that $Q_i^{\mu}$ corresponds to worst case policy and included the subscript i indicating that the Q function corresponds to  source$_i$.
> > * Updated theorem 3.8 to state M-Q-M.
> > * **Nonlinear Combination Function:** Now we state that “$Q^*_{|R_i|}$ (representing Q function with respect to reward $|R_i|$)”. Note that we explicitly refer to $R_i$ here due to the absolute sign, which distinguishes it from $Q_i$ or $Q_{R_i}$.

---

> > > ### Comment · Reviewer_KQD6 · 2026-02-21
> > > **Followup to author response**
> > >
> > > Thanks to the authors for the quick updates.
> > >
> > > I have only a couple of follow-up points from the original comments, one small, and one more substantial.
> > >
> > > 1. Regarding the meaning of “gradually increase its SBF” – good, that’s what I guessed. To avoid me or other readers guessing, I’d suggest re-phrasing to avoid any ambiguity with a single item having an increasing value: e.g., something like “To demonstrate the influence of SBF, for each evaluation domain we test with a range of SBF values.”
> > >
> > > 2. Let’s say a “real world problem” is a problem where we want a solution/good policy for its own sake, regardless of method (in contrast to a synthetic or toy problem where we care about the performance of a method more than the solution/policy). Do the authors have a “real world problem” that they would propose as a motivating example where they would use this method and expect it to work?
> > >
> > > The issue I’m getting at in this question is I don’t know if this work is intended to be a practical solution for “real world” problems, or a theoretical technique extending the boundaries of what we know is possible.
> > >
> > > As a practical work, (A) my impression is that the requirements for optimality seem hard in practice when most “real world” problems are large, and (B) I don’t see a situation where I would both expect computation to take long enough that I care to speed it up and also be willing to gamble that possibly spending a long time on the Q-M iteration pre-computation will actually be helpful.
> > >
> > > The paper doesn’t have this “real world” problem, which would seem to place it more as a theoretical work. However, much of the language of the work seems more in line with the presentation of a practical method (with some theoretical guarantees). The semi-formal treatment of sample complexity in 3.1.2 and total running time or pre-processing plus learning further push it back towards practical method: those would otherwise seem like the final piece of a complete theoretical treatment.
> > >
> > > Along which path– practical or theoretical – is a reader intended to consider this work?

---

> > > > ### Author Response · Authors · 2026-02-21
> > > > **Response to Reviewer KQD6**
> > > >
> > > > Thank you for the additional comment, and it raises a great point. Note that our motivational “real world problem” in the introduction was to adapt an autonomous driving vehicle that already knows how to drive fast or comfortably to drive both fast and comfortably.
> > > >
> > > > While that example is clearly relatable and implies strong benefits for real-world applications, we acknowledge that it is beyond the capabilities of the current method, which is limited to discrete domains and makes several assumptions. To make it practical for such domains, we must address the limitations, as well as remove or relax those assumptions. We are currently working on extending our method to continuous domains and exploring conditions we can still maintain optimality (such as ways to specify conservative models to reduce the requirement of accurate lite models).
> > > >
> > > > Hence, you are right in your implication that the main contribution is mostly theoretical, which we acknowledge, but we would also like to foreshadow its potential use in practice before we can develop the theories further.
> > > >
> > > > We have also revised the sentence in the Experimental Setup section, with the clarified wording suggested.

---

> > > > > ### Comment · Reviewer_KQD6 · 2026-02-22
> > > > > **No more comments**
> > > > >
> > > > > Thank you. I have no further questions or suggestions for minor revisions.

---

### Author Response · Authors · 2026-03-27
**Next Steps**

We have not received any indication of additional concerns from the reviewers regarding our rebuttal, and so we hope that the proposed resolutions to the previously raised issues are satisfactory (please let us know if this is not the case). We have updated our manuscript accordingly and uploaded the revised PDF. The main changes are highlighted in red and blue. We would be grateful for guidance on the next steps in the review process. We sincerely thank the reviewers and AE once again for your time and thoughtful feedback on our work.

---

### Decision · Action_Editor_QHsS · 2026-04-08

**Recommendation:** Accept as is

**Audience:**

Yes

**Audience Explanation:**

This is a paper that would be of interest to a large portion of the reinforcement learning community.

**Claims And Evidence:**

Yes

**Claims Explanation:**

This is mostly a theoretical paper and the theoretical contributions appear to be sound and correct.

The empirical evaluations are useful for supporting the theoretical contributions, but are somewhat small-scale (as raised by reviewers and agreed by authors).